# ARMOURED: ADVERSARIALLY ROBUST MODELS USING UNLABELED DATA BY REGULARIZING DIVERSITY

**Kangkang Lu**[*,1]**, Cuong Manh Nguyen**[*,1]**, Xun Xu**[1]**,**
**Yu-Jing Goh**[1,2]**, Kiran Krishnamachari**[1,2] **& Chuan-Sheng Foo**[†,1]

[1] Institute for Infocomm Research, A*STAR, Singapore
[2] National University of Singapore, Singapore
{lu_kangkang,nguyen_manh_cuong,xu_xun}@i2r.a-star.edu.sg
{yujinggoh,kirank}@u.nus.edu, foo_chuan_sheng@i2r.a-star.edu.sg

## ABSTRACT

Adversarial attacks pose a major challenge for modern deep neural networks. Recent advancements show that adversarially robust generalization requires a large amount of labeled data for training. If annotation becomes a burden, can unlabeled data help bridge the gap? In this paper, we propose ARMOURED, an adversarially robust training method based on semi-supervised learning that consists of two components. The first component applies multi-view learning to simultaneously optimize multiple independent networks and utilizes unlabeled data to enforce labeling consistency. The second component reduces adversarial transferability among the networks via diversity regularizers inspired by determinantal point processes and entropy maximization. Experimental results show that under small perturbation budgets, ARMOURED is robust against strong adaptive adversaries. Notably, ARMOURED does not rely on generating adversarial samples during training. When used in combination with adversarial training, ARMOURED yields competitive performance with the state-of-the-art adversarially-robust benchmarks on SVHN and outperforms them on CIFAR-10, while offering higher clean accuracy.

## 1 INTRODUCTION

Modern deep neural networks have met or even surpassed human-level performance on a variety of image classification tasks. However, they are vulnerable to adversarial attacks, where small, calculated perturbations in the input sample can fool a network into making unintended behaviors, e.g., misclassification. (Szegedy et al., 2014; Biggio et al., 2013). Such adversarial attacks have been found to transfer between different network architectures (Papernot et al., 2016) and are a serious concern, especially when neural networks are used in real-world applications.

As a result, much work has been done to improve the robustness of neural networks against adversarial attacks (Miller et al., 2020). Of these techniques, adversarial training (AT) (Goodfellow et al., 2015; Madry et al., 2018) is widely used and has been found to provide the most robust models in recent evaluation studies (Dong et al., 2020; Croce & Hein, 2020). Nonetheless, even models trained with AT have markedly reduced performance on adversarial samples in comparison to clean samples. Models trained with AT also have worse accuracy on clean samples when compared to models trained with standard classification losses. Schmidt et al. (2018) suggest that one reason for such reductions in model accuracy is that training adversarially robust models requires substantially more labeled data. Due to the high costs of obtaining such labeled data in real-world applications, recent work has explored semi-supervised AT-based approaches that are able to leverage unlabeled data instead (Uesato et al., 2019; Najafi et al., 2019; Zhai et al., 2019; Carmon et al., 2019).

---

[*]Equal contribution.
[†]Corresponding author.

Orthogonal to AT-based approaches that focus on training robust single models, a few works have explored the use of diversity regularization for learning adversarially robust classifiers. These works rely on encouraging ensemble diversity through regularization terms, whether on model predictions (Pang et al., 2019) or model gradients (Dabouei et al., 2020), guided by the intuition that diversity amongst the model ensemble will make it difficult for adversarial attacks to transfer between individual models, thus making the ensemble as a whole more resistant to attack.

In this work, we propose ARMOURED: Adversarially Robust MOdels using Unlabeled data by REgularizing Diversity, a novel algorithm for adversarially robust model learning that elegantly unifies semi-supervised learning and diversity regularization through a multi-view learning framework. ARMOURED applies a pseudo-label filter similar to co-training (Blum & Mitchell, 1998) to enforce consistency of different networks' predictions on the unlabeled data. In addition, we derive a regularization term inspired by determinantal point processes (DPP) (Kulesza & Taskar, 2012) that encourages the two networks to predict differently for non-target classes. Lastly, ARMOURED maximizes the entropy of the combined multi-view output on the non-target classes. We show in empirical evaluations that ARMOURED achieves state-of-the-art robustness against strong adaptive adversaries as long as the perturbations are within small $\ell_\infty$ or $\ell_2$ norm-bounded balls. Notably, unlike previous semi-supervised methods, ARMOURED does not use adversarial samples during training. When used in combination with AT, ARMOURED is competitive with the state-of-the-art methods on SVHN and outperforms them on CIFAR-10, while offering higher clean accuracy.

In summary, the major contributions of this work are as follows:

1. We propose ARMOURED, a novel semi-supervised method based on multi-view learning and diversity regularization for training adversarially robust models.

2. We perform an extensive comparison, including standard semi-supervised learning approaches in addition to methods for learning adversarially robust models.

3. We show that ARMOURED+AT achieves state-of-the-art adversarial robustness while maintaining high accuracy on clean data.

## 2 RELATED WORK

To set the stage for ARMOURED, in this section, we briefly review adversarially robust learning and semi-supervised learning - two paradigms in the literature that are related to our work.

### 2.1 ADVERSARIALLY ROBUST LEARNING

**Adversarial attacks:** We consider attacks where adversarial samples stay within a $\ell_p$ ball with fixed radius $\epsilon$ around the clean sample. In this setting, the two standard white-box attacks are the Fast Gradient Sign Method (FGSM) (Goodfellow et al., 2015) that computes a one-step perturbation that maximizes the cross entropy loss function, and Projected Gradient Descent (PGD) (Madry et al., 2018), a stronger attack that performs multiple iterations of gradient updates to maximize the loss; this may be seen as a multi-step version of FGSM. Auto-PGD attack (APGD) (Croce & Hein, 2020) is a parameter-free, budget-aware variant of PGD which aims at better convergence. However, robustness against these gradient-based attacks may give a false sense of security due to *gradient-masking*. This phenomenon happens when the defense does not produce useful gradients to generate adversarial samples (Athalye et al., 2018). Gradient-masking is known to affect PGD by preventing its convergence to the actual adversarial samples (Tramèr & Boneh, 2019). There exists gradient-based attacks such as Fast Adaptive Boundary attack (Croce & Hein, 2019) (FAB) which is invariant to rescaling, thus is unaffected by gradient-masking. FAB minimizes the perturbation norm as long as misclassification is achieved. Black box attacks that rely on random search alone without gradient information, such as Square attack (Andriushchenko et al., 2020), are also unaffected by gradient masking. Finally, AutoAttack (Croce & Hein, 2020) is a strong ensemble adversary which applies four attacks sequentially (APGD with cross entropy loss, followed by targeted APGD with difference-of-logits-ratio loss, targeted FAB, then Square).

**Adversarial training:** Adversarial training (AT) is a popular approach that performs well in practice (Dong et al., 2020). Madry et al. (2018) formulate AT as a min-max problem, where the model is trained with adversarial samples found via PGD. Variants of this method such as TRADES (Zhang

et al., 2019b) and ALP (Kannan et al., 2018) further decompose the error into natural error and boundary error for higher robustness. Zhang et al. (2019a); Wang et al. (2019) theoretically prove the convergence of AT. Two drawbacks of AT are its slow training due to adversarial example generation requiring multiple gradient computations, and the significant reduction in model accuracy on clean samples. Several recent works have focused on speeding up AT (Zhang et al., 2019a; Qin et al., 2019; Shafahi et al., 2019); ARMOURED addresses the second limitation, enabling significantly improved performance on clean samples.

**Semi-supervised adversarial training:** Schmidt et al. (2018) showed that adversarial robust generalization requires much more labeled data. To relieve the annotation burden, several semi-supervised adversarially robust learning (SSAR) methods have been developed to exploit unlabeled data instead. Uesato et al. (2019) introduced unsupervised adversarial training, a simple self-training model which optimizes a smoothness loss and a classification loss using pseudo-labels. Carmon et al. (2019) revisited the Gaussian model by Schmidt et al. (2018) and introduced robust self-training (RST), another self-training model that computes a regularization loss from unlabeled data, either via adversarial training or stability training. Zhai et al. (2019) applied a generalized virtual adversarial training to optimize the prediction stability of their model in the presence of perturbations. Najafi et al. (2019) proposed a semi-supervised extension of the distributionally robust optimization framework by Sinha et al. (2018). They replace pseudo-labels with soft-labels for unlabeled data and train them together with labeled data. It is worth noting that all of these four state-of-the-art SSAR methods apply AT in their training procedure.

**Diversity regularization:** Diversity regularization is an orthogonal direction to AT that has the potential to further improve the performance of AT. In earlier work, Pang et al. (2018) showed that for a single network, adversarial robustness can be improved when the features learned for different classes are diverse. Pang et al. (2019) further developed this concept by introducing Adaptive Promoting Diversity regularization (ADP). Given an ensemble of neural network classifiers, ADP promotes diversity among non-target predictions of the networks. ADP is inspired by determinantal point processes (Hough et al., 2006), an elegant statistical tool to model repulsive interactions among items of a fixed ground set; applications to machine learning are reviewed in (Kulesza & Taskar, 2012). Dabouei et al. (2020) enforce diversity on the gradients of individual networks in the ensemble instead of their predictions. We note that unlike ARMOURED, the methods described here are developed for the fully-supervised setting, and are not able to utilize unlabeled data.

## 2.2 SEMI-SUPERVISED LEARNING

**Semi-supervised learning:** Semi-supervised learning (SSL) is an effective strategy to learn from low-cost unlabeled data. There is considerable recent work in this practically relevant and active research area; we will not be able to cover all these works here. Existing SSL methods can be broadly categorized into three groups: consistency-based, graph-based, and generative models. Recent methods, such as Mean Teacher (Tarvainen & Valpola, 2017) and MixMatch (Berthelot et al., 2019), are consistency-based as this approach can be adapted to generic problems and have superior performance in practice. The key idea behind consistency-based methods is that model predictions on different augmentations of the same input should be consistent.

**Multi-view learning:** Multi-view learning is a SSL paradigm that is capable of representing diversity in addition to consistency. A dataset is considered to have multiple *views* when its data samples are represented by more than one set of features and each set is sufficient for the learning task. In this setting, a multi-view method assigns one modeling function to each view and jointly optimizes the functions to improve generalization performance (Zhao et al., 2017). By analyzing various multi-view algorithms, Xu et al. (2013) summarized *consensus* and *complementary* as the two underpinning principles of multi-view learning. The consensus principle states that a multi-view technique must aim at maximizing the prediction agreement on different views, similar to the consistency-based SSL methods discussed above. The complementary principle states that in order to make improvement, each view must contain some information that the other views do not carry, that the views should be sufficiently diverse. This principle has been applied to boost generalization capability in regular SSL (Qiao et al., 2018) and learning with label noise (Han et al., 2018). In this paper, we argue that multi-view complementarity also plays a critical role in improving adversarial robustness, by reducing the transferability of adversarial attacks across different views.

## 3 THE ARMOURED METHOD

In this section, we introduce ARMOURED, our proposed semi-supervised adversarially robust learning method. To utilize both labeled and unlabeled data, ARMOURED adopts a multi-view framework where multiple networks output different predictions (posterior probabilities, which we will refer to as *deep views*) on the same input image. The networks are then co-optimized by a single loss function computed on the deep views. We adhere to both the consensus and complementary principles of multi-view learning by ensuring that the deep views maximize their consensus on the target class (the ground truth class for labeled examples), but complement each other on the non-target classes. To determine a "target" class for unlabeled samples, ARMOURED applies a matching filter to pick out a target class based on agreement between views. Since our method is designed for adversarial robustness, we place a greater emphasis on the complementary principle. More concretely, we introduce two levels of complementarity: (i) among the deep views via a regularizer based on DPP and (ii) among the non-target classes via an entropy regularization applied on the combined multi-view output. Following this, we will describe ARMOURED in detail. Pseudocode detailing the training procedure is provided in Algorithm 1 in Appendix A.1.

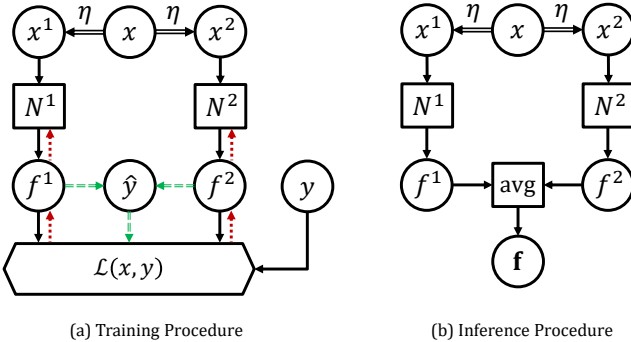

(a) Training Procedure          (b) Inference Procedure

Figure 1: ARMOURED dual-view framework: (a) training and (b) inference procedures. Solid black and dotted red arrows denote forward and backward passes, respectively; double black arrows represent image augmentation; double-dashed green arrows denote pseudo-label filter.

**Overview:** We describe the general $M$-view model. Consider a semi-supervised image classification task on input image $x$ and target label $y$ from one of $K$ classes, $y \in \{1, 2, \dots, K\}$. In each minibatch, our training data consists of a labeled set $\mathbf{L} = \{(x_i, y_i)\}_{i=1}^{n_L}$ and an unlabeled set $\mathbf{U} = \{x_i\}_{i=1}^{n_U}$. For each input image $x$, we apply random augmentations to generate $M$ different augmented images $\{x^m\}_{m=1}^{M}$. Let $\{N^m\}_{m=1}^{M}$ be architecturally similar neural networks with respective parameters $\{\theta^m\}_{m=1}^{M}$. Each network takes the corresponding augmented input and produces predictions $f^m(x) = N^m(x^m, \theta^m) \ \forall m = 1, \dots, M$. Due to the different augmentations and network parameters, each output $f^m$ can be treated as one *deep view* of the original image $x$. Finally, we compute a loss function on these deep views and backpropagate to optimize the parameters

$$\mathcal{L}(x, y) = \mathcal{L}_{\text{CE}}(x, y) + \lambda_{\text{DPP}}\mathcal{L}_{\text{DPP}}(x, y) + \lambda_{\text{NEM}}\mathcal{L}_{\text{NEM}}(x, y) \tag{1}$$

where $\lambda_{\text{DPP}}$ and $\lambda_{\text{NEM}}$ are model hyperparameters. We describe each component of the overall loss function, $\mathcal{L}_{\text{CE}}$, $\mathcal{L}_{\text{DPP}}$ and $\mathcal{L}_{\text{NEM}}$ in the following. At inference time, the $M$ outputs are combined to produce a single prediction. Since our networks possess similar learning capability, the final output is computed by averaging the deep views: $\mathbf{f}(x) = \frac{1}{M}\sum_{m=1}^{M} f^m(x)$. The detailed inference procedure is given in Algorithm 2 in Appendix A.1. Figure 1 illustrates the ARMOURED multi-view framework for the dual-view scenario.

**Cross-entropy loss ($\mathcal{L}_{\text{CE}}$) and pseudo-label filter:** For each labeled sample, we minimize the standard cross-entropy loss $\mathcal{L}_{\text{CE}}(x, y) = -\sum_{m=1}^{M} \log f_y^m(x)$. While one may train each deep view independently using only the labeled data, the fact that augmented inputs are generated from the same original image enables us to add an additional constraint – that the deep views should agree with each other even on unlabeled samples. Hence, when all $M$ networks assign the highest probability to the same class, we can be confident about their prediction on the sample. We denote such sample

as a *stable sample* and define a pseudo-label $\hat{y}$ for it as $\hat{y} = \arg\max_{k=1,\dots,K} f_k^m \quad \forall m = 1, \dots, M$. This pseudo-labeling technique has its roots in co-training (Blum & Mitchell, 1998), a multi-view technique that conforms to the consensus principle. After a stable sample is confirmed, it is treated as a labeled sample and the cross-entropy loss $\mathcal{L}_{\text{CE}}$ applies. We recompute pseudo-labels for each minibatch to avoid making incorrect pseudo-labels permanent.

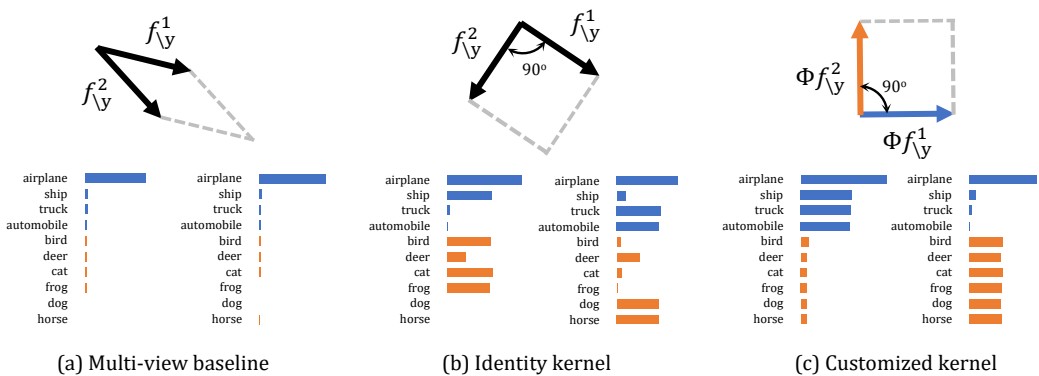

(a) Multi-view baseline        (b) Identity kernel        (c) Customized kernel

Figure 2: Three models are trained on CIFAR-10-semi-1k setup. We plot the average prediction of test samples with label "airplane". (a) ARMOURED without DPP regularization: each network predicts randomly on non-target classes. (b) ARMOURED-I: with identity matrix as kernel, network predictions on non-target classes are orthogonal. (c) ARMOURED-H: hand-crafted kernel causes a clustering effect, where each network prefers a group of classes, either vehicles or animals.

**DPP regularization ($\mathcal{L}_{\text{DPP}}$):** Suppose that the number of deep views is smaller than the number of classes, i.e., $(M < K)$. Let $F$ be the $K \times M$ matrix formed by stacking the deep views horizontally, i.e., $F = \left[ f^1, f^2, \dots, f^M \right]$. Furthermore, let $S$ be a $K \times K$ positive semidefinite kernel matrix that measures the pairwise similarity among the classes. For each sample, we extract $F_{\backslash y}$ and $S_{\backslash y}$ as the submatrices of $F$ and $S$ that correspond to the non-target classes. Let $\tilde{F}_{\backslash y}$ denote the normalized $F_{\backslash y}$ where each column is scaled to unit length. Inspired by determinantal point processes (Kulesza & Taskar, 2012), ARMOURED minimizes the following loss:
$$\mathcal{L}_{\text{DPP}}(x, y) = -\log \left[ \det \left( \tilde{F}_{\backslash y}^\top S_{\backslash y} \tilde{F}_{\backslash y} \right) \right].$$

This loss is minimized at $\tilde{F}_{\backslash y} = \tilde{F}_*$, where $\tilde{F}_*$ is the horizontal concatenation of the first $M$ dominant eigenvectors of $S_{\backslash y}$; a proof is provided in Appendix A.2. Since eigenvectors are always orthogonal, $\mathcal{L}_{\text{DPP}}$ encourages the deep views to make diverse predictions on non-target classes. If the kernel matrix is predefined, this result allows us to interpret the non-target predictions implied by the DPP regularizer. Specifically, if the kernel $S$ is constructed by a similarity measure over the classes, then a clustering effect will be observed, where similar classes are "preferred" by the same view. On the other hand, we can also inject prior knowledge or encourage desired behavior by designing a custom kernel. Exploitation of prior knowledge can be beneficial to generalization, especially when labeled training data are limited.

We note that our DPP regularizer generalizes the ensemble diversity regularizer of ADP (Pang et al., 2019), that uses the identity matrix as its kernel ($S \equiv I$). If we decompose the kernel matrix such that $S = \Phi^\top \Phi$, then our DPP regularizer is equivalent to the ADP regularizer applied on a linear transformation $\Phi \tilde{F}_{\backslash y}$ of the non-target predictions. Again, this linear transformation is another way to regulate the deep views, and can either be learned or predefined. Figure 2 illustrates the difference between the predictions from baseline model vs from ARMOURED models with different kernels. Another related work is cost-sensitive robustness (Zhang & Evans, 2018), which uses a cost matrix to weigh different adversarial transformations (attacks) among the classes. Our kernel matrix does not serve the same purpose, but the effects are similar. In our model, network preference would prevent adversarial transformations across different groups of classes.

**Non-target entropy maximization ($\mathcal{L}_{\text{NEM}}$):** Besides the multi-view diversity, we further propose an entropy regularizer that encourages larger margins among non-target classes in the final predictions $\mathbf{f}(x)$. Specifically, let $\mathbf{f}_{\backslash y}$ be the $(K - 1) \times 1$ vector of non-target predictions, and $\tilde{\mathbf{f}}_{\backslash y}$ be

the normalized vector where the elements sum up to 1. We propose to maximize the entropy defined over the normalized non-target predictions. Our entropy regularizer is therefore defined as the negative entropy $\mathcal{L}_{\text{NEM}}(x, y) = -H(\tilde{\mathbf{f}}_{\setminus y}) = \sum_{k=1}^{K-1} \tilde{\mathbf{f}}_{\setminus y} \log \tilde{\mathbf{f}}_{\setminus y}$.

This loss is minimized when all elements of $\mathbf{f}_{\setminus y}$ are equal to $\frac{1}{K-1}(1 - \mathbf{f}_y)$. Intuitively, this regularizer acts as a balancing force on the non-target predictions. It prevents ARMOURED from assigning high probability to any of the incorrect classes. We note that $\mathcal{L}_{\text{NEM}}$ differs from the entropy maximization technique adopted in Pang et al. (2019) that encourages a uniform distribution over *all* $K$ classes. Although our regularizer is similar to the complement objective proposed by Chen et al. (2019), we extend this technique to semi-supervised learning and provide more theoretical insight – we show that entropy maximization increases a lower bound on the average (logit) margin under mild assumptions (Theorem A.2 in Appendix A.3).

## 4 EXPERIMENTS

### 4.1 EXPERIMENTAL SETUP

**Dataset:** We evaluate ARMOURED on the CIFAR-10 and SVHN datasets. We use the official train/test splits (50k/10k labeled samples) for CIFAR-10 (Krizhevsky et al., 2009) and reserve 5k samples from the training samples for a validation set. In our semi-supervised setup, the label budget is either 1k or 4k; remaining samples from training set are treated as unlabeled samples. For the SVHN dataset (Netzer et al., 2011), our train/validation/test split is 65,932 / 7,325 / 26,032 samples. We use only 1k samples as the label budget in our semi-supervised setup for SVHN. For simplicity, we will refer to our setup as "Dataset-semi-budget", e.g., CIFAR-10-semi-4k, SVHN-semi-1k.

**Adversarial attacks:** To evaluate robustness, we apply the following adversaries: (i) Fast Gradient Sign Method attack (Goodfellow et al., 2015) (FGSM), (ii) Projected Gradient Descent attack (Madry et al., 2018) (PGD) with random initialization and (iv) AutoAttack (Croce & Hein, 2020). For $\ell_\infty$ attacks, the default perturbation budget is $\epsilon = 8/255$; for $\ell_2$ attacks, $\epsilon = 0.5$.

**Backbone network and training:** To enable fair comparison, the same Wide ResNet (Oliver et al., 2018) backbone is used for all methods. Specifically, we implement "WRN-28-2" with depth 28 and width 2 along with batch normalization, leaky ReLU activation and Adam optimizer. We train each method for 600 epochs on CIFAR-10-semi-4k and SVHN-semi-1k. Learning rate is decayed by a factor of 0.2 after the first 400k iterations.

**AT wrapper for SSL:** We notice that many concepts in SSL, such as multi-view diversity or consistency, are orthogonal to AT, and that successful defenses against large-perturbation attacks always rely on AT (Croce & Hein, 2020). Therefore, we hope to combine the best of both worlds by implementing AT as a wrapper method for SSL. Algorithm 3 in Appendix A.1 describes our **Method+AT** wrapper, which consists of three main steps. First, for each batch of semi-supervised data, we apply the inference procedure of the **Method** (e.g., Algorithm 2 of ARMOURED) to generate pseudo-labels for unlabeled data. Second, for each input sample in the batch, we compute its adversarial sample using either the true label (if the sample is labeled) or the pseudo-label (if the sample is unlabeled). Third, we execute the training procedure of the **Method** (e.g., Algorithm 1 of ARMOURED) using the adversarial samples and the original labels. The pseudo-labels computed from the first step are now dropped, so that the training is still semi-supervised. We note that this wrapper algorithm resembles RST (Carmon et al., 2019).

**ARMOURED variants:** We design three variants based on the dual-view model shown in Figure 1 that differ only in choice of diversity kernel. ARMOURED-I is our standard model that uses the **I**dentity matrix as its diversity kernel. ARMOURED-H uses a **H**and-crafted binary matrix intended to group the classes into two predefined clusters. On CIFAR-10, these are "vehicles" (airplane, ship, truck, automobile) vs. "animals" (bird, cat, deer, dog, frog, horse). On SVHN, we split the digits into "simple & edgy" (0, 1, 2, 4, 7) vs. "curvy & loopy" (3, 5, 6, 8, 9). The third variant is ARMOURED-F, which uses a learnable **F**eature-based kernel. From a pre-trained SSL model, we first compute the adversarial samples corresponding to the labeled training samples. Then, for each class, we extract feature vectors by averaging over the adversarial samples associated with the class. Finally, we combine the feature vectors into a matrix $B$ and compute a kernel $S = B^\top B$. More details of the kernels are provided in Appendix A.4. In our experiments, we evaluate the following

four variants: ARMOURED-I+AT, ARMOURED-H+AT, ARMOURED-F+AT and ARMOURED-F (trained without AT). For the AT wrapper, we apply a 7-step PGD $\ell_\infty$ attack with total $\epsilon = 8/255$ (for CIFAR-10), $\epsilon = 4/255$ (for SVHN) and step size of $\epsilon/4$.

**Comparison benchmarks:** We test the proposed method against a wide range of state-of-the-art SSL and SSAR benchmarks: Mean Teacher (MT) (Tarvainen & Valpola, 2017), MixMatch (Berthelot et al., 2019), RST (Carmon et al., 2019) (RST has two variants, we implemented $\text{RST}_\text{adv}$), and the method of Zhai et al. (2019) that we denote as ARG. In addition, we combine MT with adversarial training (MT+AT) using the wrapper Algorithm 3 in Appendix A.1. To the best of our knowledge, this is the first time MT+AT has been evaluated for adversarial robustness. For AT-based methods (RST, ARG), we use a 7-step PGD $\ell_\infty$ attack in their AT phase, similar to MT+AT and ARMOURED+AT.

## 4.2 RESULTS[1]

**Results on CIFAR-10 (Table 1, Figure 3):** On clean data, MixMatch yields the best performance, while ARMOURED-F surpasses all methods trained with AT by large margins (18%-26%) and is even better than MT – a SSL method. ARMOURED variants demonstrate substantially higher clean performance over the SSAR benchmarks. Under standard FGSM and PGD attacks, the most robust defense is still ARMOURED-F, followed by its AT-based variants with accuracy drops of 2%-5%. Other methods shows larger gaps: 25%-35% for SSAR and 10%-50% for SSL benchmarks. We note that the improvements by ARMOURED are not due to gradient masking (see Appendix B). Under AutoAttack, ARMOURED-F is no longer robust, instead, ARMOURED+AT variants are more resilient. ARMOURED-F+AT is the best defense, outperforming ARG by 5.23% for $\ell_\infty$ and 9.85% for $\ell_2$ attacks. We also notice that the two best defenses against AutoAttack are trained with the hand-crafted and feature kernels. The former requires only human knowledge while the latter just needs additional computing resources, giving our method flexible ways to boost adversarial robustness with or without prior knowledge.

In Figure 3, we plot the robust accuracy against AutoAttack as the perturbation budget $\epsilon$ gradually increases. ARMOURED-F obtains highest accuracy on clean data as well as for small perturbation budgets, but its accuracy drops rapidly as $\epsilon$ is increased. Meanwhile, the ARMOURED+AT variants are able to achieve a better trade-off between clean accuracy and robustness.

Table 1: Benchmark results on CIFAR-10-semi-4k

| Method | Clean | FGSM $\ell_\infty$ | PGD $\ell_\infty$ | PGD $\ell_2$ | AutoAttack $\ell_\infty$ | AutoAttack $\ell_2$ |
|---|---|---|---|---|---|---|
| MT | $84.55 \pm 0.64$ | $14.00 \pm 1.70$ | $0.03 \pm 0.02$ | $19.55 \pm 0.26$ | $0.01 \pm 0.01$ | $0.46 \pm 0.02$ |
| MixMatch | $\mathbf{89.95 \pm 0.96}$ | $57.42 \pm 2.78$ | $0.25 \pm 0.14$ | $9.10 \pm 0.82$ | $0.00 \pm 0.00$ | $0.01 \pm 0.01$ |
| RST | $58.81 \pm 0.28$ | $33.80 \pm 0.20$ | $31.28 \pm 0.26$ | $43.70 \pm 0.33$ | $25.38 \pm 0.15$ | $39.79 \pm 0.60$ |
| ARG | $67.07 \pm 0.22$ | $36.65 \pm 1.79$ | $32.12 \pm 1.74$ | $42.95 \pm 0.82$ | $30.01 \pm 1.85$ | $40.86 \pm 0.69$ |
| MT+AT | $64.77 \pm 0.28$ | $30.50 \pm 3.16$ | $25.76 \pm 3.16$ | $41.34 \pm 1.76$ | $24.28 \pm 2.90$ | $39.93 \pm 1.63$ |
| ARMOURED-F | $84.90 \pm 0.22$ | $\mathbf{68.27 \pm 1.53}$ | $\mathbf{56.42 \pm 3.93}$ | $\mathbf{67.45 \pm 2.76}$ | $8.70 \pm 1.27$ | $25.80 \pm 2.50$ |
| ARMOURED-I+AT | $77.73 \pm 0.11$ | $63.56 \pm 1.71$ | $54.22 \pm 3.68$ | $61.95 \pm 2.11$ | $29.22 \pm 3.18$ | $44.44 \pm 3.15$ |
| ARMOURED-H+AT | $76.74 \pm 0.73$ | $63.41 \pm 1.94$ | $54.94 \pm 4.30$ | $61.89 \pm 2.74$ | $31.88 \pm 7.19$ | $46.29 \pm 5.84$ |
| ARMOURED-F+AT | $76.76 \pm 1.60$ | $64.05 \pm 3.00$ | $55.12 \pm 4.90$ | $61.93 \pm 3.06$ | $\mathbf{35.24 \pm 4.56}$ | $\mathbf{49.78 \pm 3.80}$ |

**Results on SVHN (Table 2):** On clean test samples, ARMOURED-F yields the best performance, with a small improvement over the second best competing method. Against FGSM and PGD attacks, even the worst ARMOURED variant is more robust than MT+AT (the best benchmark) by 13%-20%. Under AutoAttack, ARMOURED-H+AT falls behind MT+AT and ARG by a significant gap of 15% under $\ell_\infty$ attacks, while outperforming them by 4%-11% under $\ell_2$ attacks. Overall, ARMOURED shows competitive performance compared to state-of-the-art SSL and SSAR benchmarks. Results on CIFAR-10 and SVHN suggest that MT+AT is a strong defense.

**Ablation study (Table 3):** We perform an ablation study to investigate the contribution of each component to the performance of ARMOURED-F+AT agaisnt AutoAttack. First, we remove both DPP and entropy regularization terms from the total loss in equation (1). This model, denoted as **w/o**

---

[1]Each result contains mean and standard deviation statistics computed from three independent runs with different random data seeds (for selecting labeled samples).

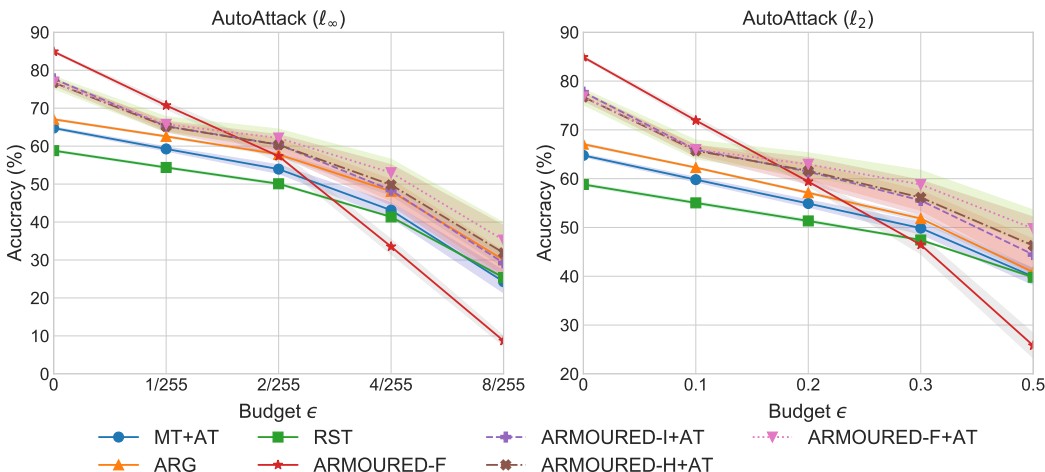

Figure 3: Robustness against AutoAttack vs. perturbation budget $\epsilon$ on CIFAR-10-semi-4k.

Table 2: Benchmark results on SVHN-semi-1k

| Method | Clean | FGSM $\ell_\infty$ | PGD $\ell_\infty$ | PGD $\ell_2$ | AutoAttack $\ell_\infty$ | AutoAttack $\ell_2$ |
|---|---|---|---|---|---|---|
| MT | $93.73 \pm 0.96$ | $24.44 \pm 2.13$ | $4.14 \pm 0.81$ | $34.64 \pm 3.12$ | $1.50 \pm 0.37$ | $12.43 \pm 1.53$ |
| MixMatch | $91.35 \pm 1.00$ | $43.75 \pm 23.50$ | $0.05 \pm 0.05$ | $3.72 \pm 3.31$ | $0.00 \pm 0.00$ | $0.21 \pm 0.20$ |
| RST | $55.67 \pm 1.79$ | $30.40 \pm 1.30$ | $25.99 \pm 1.79$ | $33.71 \pm 3.32$ | $17.33 \pm 1.94$ | $25.08 \pm 3.79$ |
| ARG | $91.66 \pm 0.51$ | $58.22 \pm 0.95$ | $44.42 \pm 1.11$ | $44.12 \pm 1.07$ | $\mathbf{39.50 \pm 1.11}$ | $34.19 \pm 0.93$ |
| MT+AT | $92.71 \pm 0.22$ | $58.45 \pm 1.02$ | $44.70 \pm 0.42$ | $50.42 \pm 1.02$ | $38.06 \pm 0.11$ | $41.71 \pm 0.93$ |
| ARMOURED-F | $\mathbf{93.93 \pm 0.79}$ | $72.11 \pm 1.67$ | $57.91 \pm 5.08$ | $\mathbf{73.44 \pm 3.18}$ | $7.88 \pm 0.87$ | $24.62 \pm 1.38$ |
| ARMOURED-I+AT | $92.01 \pm 0.44$ | $74.22 \pm 1.32$ | $62.04 \pm 4.93$ | $71.38 \pm 2.28$ | $22.04 \pm 3.04$ | $41.57 \pm 2.45$ |
| ARMOURED-H+AT | $92.53 \pm 0.35$ | $73.54 \pm 1.58$ | $61.41 \pm 1.14$ | $70.53 \pm 0.71$ | $24.04 \pm 1.30$ | $\mathbf{45.58 \pm 2.39}$ |
| ARMOURED-F+AT | $92.44 \pm 0.64$ | $\mathbf{74.78 \pm 4.37}$ | $\mathbf{62.10 \pm 8.39}$ | $71.37 \pm 6.32$ | $23.35 \pm 3.22$ | $44.41 \pm 3.20$ |

($\mathcal{L}_{\mathrm{DPP}} + \mathcal{L}_{\mathrm{NEM}}$), performs relatively well on clean data, but its performance suffers under attacks, dropping by 26% for $\ell_\infty$ and by 22% for $\ell_2$ attacks. We then keep the term $\mathcal{L}_{\mathrm{NEM}}$, but remove the diversity regularizer from the loss function[2]. This model – **w/o** $\mathcal{L}_{\mathrm{DPP}}$ – performs worse than the complete model by 1%. We conclude that the entropy regularizer plays a more vital role than the DPP regularizer. Besides, we train ARMOURED-F+AT using only the 4k labeled samples and call this model **w/o Unlabeled**. Its poor performance reinforce the importance of unlabeled data towards improving adversarial robustness. Finally, we include ARMOURED-F (trained without AT), which performs very well on clean data but fails against AutoAttack. Additional results are provided in Appendix B.

**Visualization of learned representations (Figure 4)**: On CIFAR-10 test samples, we visualize the feature embeddings (extracted from the last layer of WRN-28-2 before the linear layer) learned by the four models in our ablation study. On clean test samples, ARMOURED-F produces the best embeddings. On adversarial samples, we observe gradual improvements in the representations, starting from (a) no diversity regularization to (b) diversity on only labeled samples, with network $N^1$ showing well-defined clusters; then (c) diversity on whole training set with better cluster separation and (d) combining diversity with AT, where clusters are less contaminated under attacks.

## 5 CONCLUSION

In this work, we presented ARMOURED, a novel method for learning adversarially robust models that unifies semi-supervised learning and diversity regularization in a multi-view framework. AR-MOURED alone is robust against standard white-box attacks as well as strong adaptive attacks with

---

[2]The DPP regularization term cannot function properly without entropy regularization, due to a trivial optimum at the one-hot vector $\mathbf{1}_y$, as shown by Pang et al. (2019). Hence, we must keep $\mathcal{L}_{\mathrm{NEM}}$.

Table 3: Ablation study on CIFAR-10-semi-4k

| Method | Clean | AutoAttack $\ell_\infty$ | AutoAttack $\ell_2$ |
|---|---|---|---|
| **w/o** ($\mathcal{L}_{\text{DPP}} + \mathcal{L}_{\text{NEM}}$) | $75.60 \pm 0.44$ | $9.27 \pm 0.28$ | $27.32 \pm 0.85$ |
| **w/o** $\mathcal{L}_{\text{DPP}}$ | $77.44 \pm 0.85$ | $34.02 \pm 2.80$ | $48.52 \pm 1.79$ |
| **w/o Unlabeled** | $72.50 \pm 0.21$ | $10.26 \pm 1.57$ | $25.23 \pm 0.59$ |
| ARMOURED-F | $\mathbf{84.90 \pm 0.22}$ | $8.70 \pm 1.27$ | $25.80 \pm 2.50$ |
| ARMOURED-F+AT | $76.76 \pm 1.60$ | $\mathbf{35.24 \pm 4.56}$ | $\mathbf{49.78 \pm 3.80}$ |

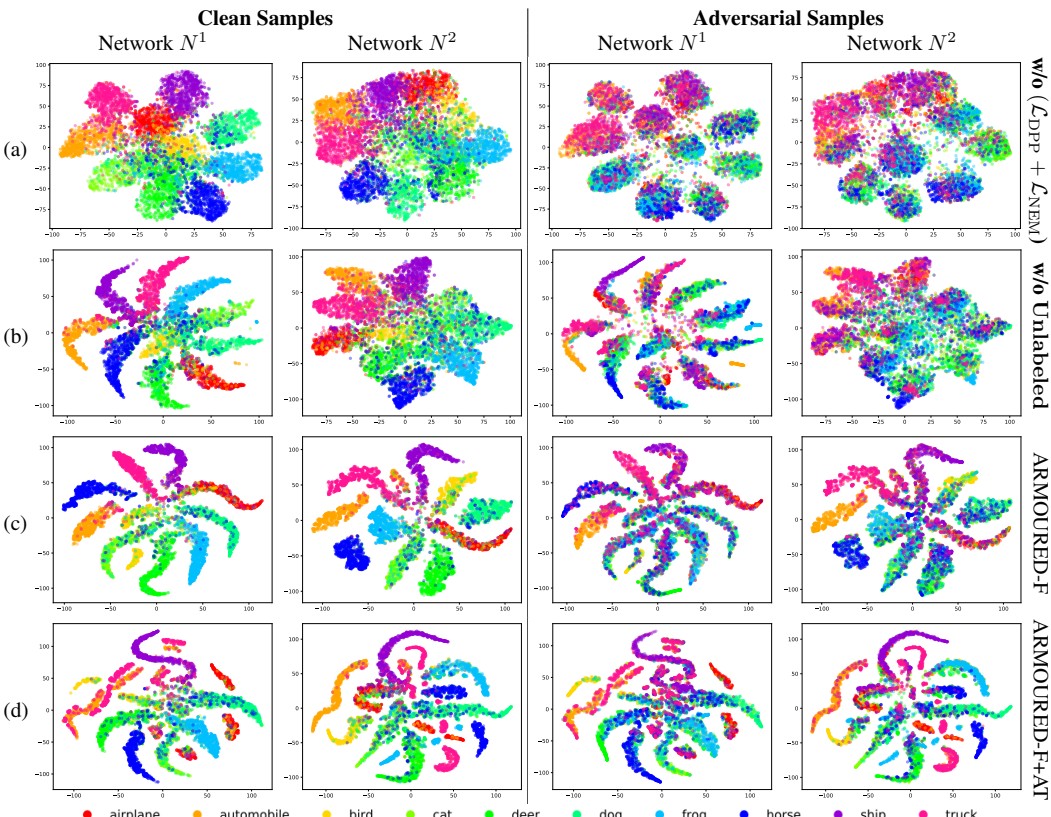

Figure 4: t-SNE plots of feature embeddings from CIFAR-10 test samples generated by ablation models: (a) **w/o** ($\mathcal{L}_{\text{DPP}} + \mathcal{L}_{\text{NEM}}$), (b) **w/o Unlabeled**, (c) ARMOURED-F and (d) ARMOURED-F+AT. For each of the 8 network/method pairs, the clean and adversarial samples are processed together in a single t-SNE run. Adversarial samples are generated with PGD-$\ell_\infty$ ($\epsilon = 8/255$). From (a) to (d), the embeddings of adversarial samples are progressively enhanced, while (c) yields the best representations on clean data.

small perturbation budgets. When combined with adversarial training, ARMOURED demonstrates much better robustness against a wider range of perturbation budgets. Additionally, ARMOURED improves clean accuracy when compared with state-of-the-art semi-supervised adversarial training methods. The empirical performance of ARMOURED+AT suggests that it is possible to learn adversarially robust models while upholding a reasonable accuracy on clean samples. Extending this method to exploit more than two views or alternative custom kernels for the DPP regularizer could result in further performance gains.

ACKNOWLEDGMENTS

This work is supported by the DSO National Laboratories of Singapore. The authors would like to thank the DSO project team, in particular Dr. Loo Nin Teow and Dr. Bingquan Shen for valuable discussions on adversarial robustness.

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

APPENDICES

# A   ARMOURED METHOD

## A.1   DETAILED PSEUDOCODES

---

**Algorithm 1:** ARMOURED Minibatch Training Procedure

---

**Input:** Labeled samples $\mathbf{L} = \{(x_i, y_i)\}_{i=1}^{n_L}$; unlabeled samples $\mathbf{U} = \{x_i\}_{i=1}^{n_U}$; kernel matrix $S$; random augmentation $\eta(x)$; hyperparameters $(\lambda_{\text{DPP}}, \lambda_{\text{NEM}})$
**Output:** Networks $\{N^m\}_{m=1}^M$ with updated parameters $\{\theta^m\}_{m=1}^M$

**for** $i = 1, \ldots, n_U$ **do**
    **for** $m = 1, \ldots, M$ **do**
        $x_i^m = \eta(x_i)$ // random augmentation
        $f^m(x_i) = N^m(x_i^m, \theta^m)$ // forward pass
    **end**
    **if** $\hat{y}_i = \arg\max_{k=1,\ldots,K} f_k^m(x_i) \, \forall m = 1, \ldots, M$ **then**
        add $(x_i, \hat{y}_i)$ to $\mathbf{L}$
        remove $x_i$ from $\mathbf{U}$
    **end**
**end**
**for** $i = 1, \ldots, n_l$ **do**
    **for** $m = 1, \ldots, M$ **do**
        $x_i^m = \eta(x_i)$ // random augmentation
        $f^m(x_i) = N^m(x_i^m, \theta^m)$ // forward pass
    **end**
    $\mathcal{L}(x_i, y_i) = \mathcal{L}_{\text{CE}}(x_i, y_i) + \lambda_{\text{DPP}}\mathcal{L}_{\text{DPP}}(x_i, y_i) + \lambda_{\text{NEM}}\mathcal{L}_{\text{NEM}}(x_i, y_i)$ // sample loss
**end**
$\mathcal{L} = \sum_{i=1}^{n_L} \mathcal{L}(x_i, y_i)$ // batch loss
backpropagate $\mathcal{L}$ to optimize $\{\theta^m\}_{m=1}^M$ // backward pass

---

---

**Algorithm 2:** ARMOURED Inference Procedure

---

**Input:** Sample $x$; networks $\{N^m\}_{m=1}^M$ with parameters $\{\theta^m\}_{m=1}^M$; random augmentation $\eta(x)$
**Output:** Posterior output $\mathbf{f}(x)$; predicted label $\hat{y}$

**for** $m = 1, \ldots, M$ **do**
    $x^m = \eta(x)$ // random augmentation
    $f^m(x) = N^m(x^m, \theta^m)$ // forward pass
**end**
$\mathbf{f}(x) = \frac{1}{M}\sum_{m=1}^M f^m(x)$ // posterior output
$\hat{y} = \arg\max_{k=1,\ldots,K} \mathbf{f}_k(x)$ // predicted label

---

## A.2   OPTIMA OF DPP REGULARIZER

For simplicity, we find the maximum of the exponential of negative loss $\mathcal{L}_{\text{DPP}}(x, y)$, defined as

$$Q(x, y) = \exp\left[-\mathcal{L}_{\text{DPP}}(x, y)\right] = \det\left(\tilde{F}_{\backslash y}^\top S_{\backslash y} \tilde{F}_{\backslash y}\right) \tag{2}$$

Since $S_{\backslash y}$ is a principal submatrix of $S$, it is also positive semidefinite. We can decompose $S_{\backslash y}$ as follows: $S_{\backslash y} = VDV^\top$, where $V$ is a square matrix whose $k$-th column is the eigenvector $v_k$ of $S_{\backslash y}$, and $D$ is a diagonal matrix whose $(k, k)$-th element $\lambda_k$ is the $k$-th largest eigenvalue of $S_{\backslash y}$.

---

**Algorithm 3:** Method+AT Minibatch Training Procedure

---

**Input:** Labeled samples $\mathbf{L} = \{(x_i, y_i)\}_{i=1}^{n_L}$; unlabeled samples $\mathbf{U} = \{x_i\}_{i=1}^{n_U}$;
SSL technique **Method** with hyperparameters $\Omega_{\mathbf{Method}}$; adversarial attack $\pi(x, y)$
**Output:** **Method** model with updated parameters

create $\mathbf{L_{adv}} = \emptyset$ and $\mathbf{U_{adv}} = \emptyset$ // new empty sets
**for** $i = 1, \ldots, n_L$ **do**
    $z_i = \pi(x_i, y_i)$ // adversarial sample
    add $(z_i, y_i)$ to $\mathbf{L_{adv}}$
**end**
**for** $i = 1, \ldots, n_U$ **do**
    apply inference procedure of **Method** on $x_i$ to generate pseudo-label $\hat{y}_i$
    $z_i = \pi(x_i, \hat{y}_i)$ // adversarial sample
    add $(z_i)$ to $\mathbf{U_{adv}}$
**end**
execute training procedure of **Method** with inputs: $\mathbf{L_{adv}}$; $\mathbf{U_{adv}}$; $\Omega_{\mathbf{Method}}$

---

The gradient of $Q$ with respect to $\tilde{F}_{\backslash y}$ is given by Petersen & Pedersen (2012) as

$$\frac{\partial Q}{\partial \tilde{F}_{\backslash y}} = 2 \det(\tilde{F}_{\backslash y}^\top S_{\backslash y} \tilde{F}_{\backslash y}) S_{\backslash y} \tilde{F}_{\backslash y} (\tilde{F}_{\backslash y}^\top S_{\backslash y} \tilde{F}_{\backslash y})^{-1} \tag{3}$$

Let $\tilde{F}_*$ be the horizontal concatenation of the first $M$ eigenvectors, i.e., $\tilde{F}_* = [v_1, v_2, \ldots, v_M]$. Notice that $\tilde{F}_*^\top S_{\backslash y} \tilde{F}_* = D_M$, where $D_M$ is the $M \times M$ leading principal submatrix of $D$. We evaluate the gradient at $\tilde{F}_*$ as follows

$$\left. \frac{\partial Q}{\partial \tilde{F}_{\backslash y}} \right|_{\tilde{F}_*} = 2 \det(\tilde{F}_*^\top S_{\backslash y} \tilde{F}_*) S_{\backslash y} \tilde{F}_* (\tilde{F}_*^\top S_{\backslash y} \tilde{F}_*)^{-1} \tag{4}$$

$$= 2 \det(D_M) S_{\backslash y} \tilde{F}_* D_M^{-1} \tag{5}$$

$$= 2 \det(D_M) D_M \tilde{F}_* D_M^{-1} \tag{6}$$

$$= 2 \det(D_M) \tilde{F}_* \tag{7}$$

Interestingly, since $D_M$ is a diagonal matrix, $\det(D_M)$ equals the product of the first $M$ eigenvalues of $S_{\backslash y}$. This product is also nonnegative because $S_{\backslash y}$ is positive semidefinite. Therefore, the gradient at $\tilde{F}_*$ is a nonnegative scaling of $\tilde{F}_*$ itself. Since $\tilde{F}_*$ is normalized to unit length, adding this gradient does not update it any further, i.e., the angular gradient at $\tilde{F}_*$ is zero. As shown by Cover & Thomas (1988), given a fixed positive semidefinite kernel, the determinant in equation (2) is a concave function of $\tilde{F}_{\backslash y}$. Thus, $\tilde{F}_*$ is a maximum of $Q$.

Note that $\tilde{F}_*$ is not the only maximum. Let $R$ be a $M \times M$ orthogonal matrix, so that $\tilde{F}_* R$ is a rotation of $\tilde{F}_*$. Then, $\tilde{F}_* R$ is also a maximum of $Q$, because

$$(\tilde{F}_* R)^\top S_{\backslash y} (\tilde{F}_* R) = R^\top (\tilde{F}_*^\top S_{\backslash y} \tilde{F}_*) R = R^\top D_M R = D_M R^\top R = D_M = \tilde{F}_*^\top S_{\backslash y} \tilde{F}_* \tag{8}$$

This means that a family of maxima exists for $Q$, which includes $\tilde{F}_*$ and its orthogonal transformations in the $M$-dimensional subspace spanned by $\tilde{F}_*$.

For example, when $M = 2$, objective $Q$ is maximized at $\tilde{F}_* = [v_1, v_2]$

$$\det\left( \begin{bmatrix} v_1^\top \\ v_2^\top \end{bmatrix} S_{\backslash y} \begin{bmatrix} v_1 & v_2 \end{bmatrix} \right) = \det\left( \begin{bmatrix} v_1^\top S_{\backslash y} v_1 & v_1^\top S_{\backslash y} v_2 \\ v_2^\top S_{\backslash y} v_1 & v_2^\top S_{\backslash y} v_2 \end{bmatrix} \right) = \det\left( \begin{bmatrix} \lambda_1 & 0 \\ 0 & \lambda_2 \end{bmatrix} \right) = \lambda_1 \lambda_2 \tag{9}$$

Any rotation of $(v_1, v_2)$ in the 2-dimensional plane spanned by them is also a maximum.

A.3 ANALYSIS OF NON-TARGET ENTROPY MAXIMIZATION

For ease of exposition, we denote $g(x)$ as the unnormalized logits of $\mathbf{f}(x)$, the Lipschitz constant $L_N$ as the scalar satisfying,

$$||g(x) - g(x + \epsilon)||_2 \leq L_N ||\epsilon||_2 \tag{10}$$

The *guarded adversarial area* (Tsuzuku et al., 2018) is defined as the hypersphere satisfying the following condition, where $c$ is the largest perturbation radius measured in $\ell_p$ distance

$$\forall \epsilon : ||\epsilon||_p \leq c \Rightarrow \mathbf{f}_y(x + \epsilon) \geq \max \mathbf{f}_{\setminus y}(x + \epsilon) \tag{11}$$

The max/average logit gap is the gap bewteen target class logit and maxmimal/average non target class logit,

$$maxgap(x) = g_y(x) - \max_{k \neq y} g_k(x), \quad avggap(x) = g_y(x) - \underset{k \neq y}{\mathrm{avg}}\, g_k(x) \tag{12}$$

We start by introducing the following lemma which is related to Proposition 1 of Tsuzuku et al. (2018).

**Lemma A.1** *For any adversarial perturbation $\epsilon$ smaller than the logit gap divided by the Liptschitz constant, it is guaranteed the class prediction does not change.*

*Proof.* Lemma A.1 can be written as the following,

$$maxgap(x) = g_y(x) - \max_{k \neq y} g_k(x) \geq \sqrt{2} L_N ||\epsilon|| \Rightarrow g_y(x + \epsilon) - \max_{k \neq y}\{(g_k(x + \epsilon)\} \geq 0 \tag{13}$$

A proof that Lemma A.1 is the same with the proof for Proposition 1 of Tsuzuku et al. (2018).

This lemma suggests that it is possible to increase the robustness, the guarded adversarial area $\epsilon$, by either decreasing the Lipschitz constant and/or increasing the logit gap. It is often acknowledged, as with the analysis in Tsuzuku et al. (2018), that the Lipschitz constant for large neural network is very hard to quantify. Instead we find it is easier to enlarge the logit gap by non-target entropy maximization and reveal a relation between them as follows,

**Theorem A.2** *The non-target entropy $H(\tilde{\mathbf{f}}_{\setminus y})$ is a lower bound of average logit gap plus a constant.*

The entropy maximization term will encourage a uniform distribution over non-target classes, i.e. $maxgap(x) \approx avggap(x)$. By referring to Lemma A.1, this theorem suggests maximizing non-target entropy $H(\tilde{\mathbf{f}}_{\setminus y})$ leads to higher guarded adversarial attack area $\epsilon$. As result, the overall robustness to adversarial attack is improved by introducing the additional non-target entropy maximization loss.

*Proof.* We first write the theorem to prove as the following:

$$H(\tilde{\mathbf{f}}_{\setminus y}) \leq g_y(x) - \underset{k \neq y}{\mathrm{avg}}\{(g_k(x)\} + C \tag{14}$$

Before we provide the proof, we introduce the following two lemmas and make a mild assumption:

**Lemma A.3** *LogSumExp is a smooth approximation to and upper bounded by the maximum function plus a constant.*

$$\log \sum_{k \neq y} \exp g_k \leq \max_{k \neq y} g_k + \log(K - 1) \tag{15}$$

*Proof.* We relax the summation with maximization and arrive at the following inequalities.

$$\log \sum_{k \neq y} \exp g_k \leq \log((K - 1) \exp(\max_{k \neq y} g_k))$$

$$= \max_{k \neq y} g_k + \log(K - 1) \tag{16}$$

**Lemma A.4** *The following inequality holds for real number vector $g$ of length $K$.*

$$\underset{k}{\mathrm{avg}}\, g_k \leq \sum_k \frac{g_k \exp g_k}{\sum_k \exp g_k} \tag{17}$$

*Proof.* W.l.o.g. we assume $g_k$ is in descending order, i.e. $\forall i < j, \quad g_i \geq g_j$. The proof is rewritten as,

$$g_1(\exp g_1 + \cdots \exp g_K) + \cdots g_K(\exp g_1 + \cdots \exp g_K) \leq K g_1 \exp g_1 + \cdots K g_K \exp g_K \quad (18)$$

The difference between RHS and LHS is written as,

$$RHS - LHS = (\exp g_1 - \exp g_2)(g_1 - g_2) + (\exp g_1 - \exp g_3)(g_1 - g_3) + \quad (19)$$
$$\cdots + (\exp g_1 - \exp g_K)(g_1 - g_K) + \quad (20)$$
$$(\exp g_2 - \exp g_3)(g_2 - g_3) + (\exp g_2 - \exp g_4)(g_2 - g_4) + \quad (21)$$
$$\cdots + (\exp g_2 - \exp g_N)(g_2 - g_N) + \quad (22)$$
$$\cdots + (\exp g_{K-1} - \exp g_K)(g_{K-1} - g_K) \quad (23)$$

Obviously, $RHS - LHS$ is non-negative, thus the inequality holds.

**Assumption A.5** *Assume the clean samples are mostly correctly classified.*

$$\max_{k \neq y} g_k(x) \leq g_y(x) \quad (24)$$

Given the fact that we can achieve relatively high classification accuracy on clean samples, the assumption is realistic in most cases.

Now we prove the inequality for equation (14) holds.

$$H(\tilde{\mathbf{f}}_{\backslash y}) = -\sum_{k \neq y} \frac{\exp g_k}{\sum\limits_{k \neq y} \exp g_k} \log \frac{\exp g_k}{\sum\limits_{k \neq y} \exp g_k} \quad (25)$$

$$= \sum_{k \neq y} \frac{\exp g_k}{\sum\limits_{k \neq y} \exp g_k} (\log \sum_{k \neq y} \exp g_k - g_k) \quad (26)$$

$$\leq \sum_{k \neq y} \frac{\exp g_k}{\sum\limits_{k \neq y} \exp g_k} (\max_{k \neq y} g_k + \log(K - 1) - g_k) \quad (27)$$

$$\leq \sum_{k \neq y} \frac{\exp g_k}{\sum\limits_{k \neq y} \exp g_k} (g_y + \log(K - 1) - g_k) \quad (28)$$

$$\leq \sum_{k \neq y} \frac{\exp g_k}{\sum\limits_{k \neq y} \exp g_k} (g_y + \log(K - 1)) - \underset{k \neq y}{\text{avg}}\, g_k \quad (29)$$

$$\leq (g_y - \underset{k \neq y}{\text{avg}}\, g_k + \log(K - 1)) \quad (30)$$

## A.4 IMPLEMENTATION DETAILS

**Hyperparameters:** We fine tune $\lambda_{\text{DPP}}$ and $\lambda_{\text{NEM}}$ with ARMOURED-I+AT model trained on CIFAR10 and SVHN individually. The tuning ranges are as follows: $\lambda_{\text{DPP}} \in [0.25, 0.5, 1]$ and $\lambda_{\text{NEM}} \in [1, 2, 4]$. Each model is trained with one seed and is evaluated on the standard validation set (5k labeled samples for CIFAR-10 and 7325 labeled samples for SVHN). Please see Table A.4 and Table A.5 for the numerical results. After tuning, we decide to apply $(\lambda_{\text{DPP}}, \lambda_{\text{NEM}}) = (1, 1)$ for SVHN and $(\lambda_{\text{DPP}}, \lambda_{\text{NEM}}) = (1, 0.5)$ for CIFAR-10.

**Random augmentations:** Regarding the random augmentations $\eta(x)$, we apply translations and horizontal flips on CIFAR-10 images and apply only random translations for SVHN images.

**Feature-based kernel:** We learn the feature-based kernel following the steps below. The learned kernels are plotted in Figure A.5.

1. Train a MT+AT model using both labeled and unlabeled training data.
2. Using the teacher model, feed forward the adversarial samples generated from labeled training data. Extract feature vectors from the last layer of WRN-28-2 before the linear layer.

3. For each class, compute the average feature vectors: $b_1, b_2, \ldots b_K$.

4. Normalize each feature vector by its L-2 norm.

5. Generate the feature matrix $B = [b_1 b_2 \ldots b_K]$ and compute the kernel as $S = B^\top B$.

6. Normalize $S$ by its largest eigenvalue (equivalent to L-2 normalization).

Table A.4: Fine-tuning of $\lambda_{\mathrm{DPP}}$ and $\lambda_{\mathrm{NEM}}$ on CIFAR10-semi-4k (reporting validation accuracy)

| | Autoattack $\ell_\infty$ | | | | Autoattack $\ell_2$ | | |
|---|---|---|---|---|---|---|---|
| $\lambda_{\mathrm{DPP}}$ \ $\lambda_{\mathrm{NEM}}$ | 1 | 2 | 4 | $\lambda_{\mathrm{DPP}}$ \ $\lambda_{\mathrm{NEM}}$ | 1 | 2 | 4 |
| 0.25 | 28.50 | 25.67 | 6.48 | 0.25 | 44.08 | 41.56 | 20.02 |
| 0.5 | 32.60 | 26.25 | 6.45 | 0.5 | 48.08 | 41.21 | 21.04 |
| 1 | 32.45 | 27.87 | 7.83 | 1 | 46.07 | 41.29 | 22.80 |

Table A.5: Fine-tuning of $\lambda_{\mathrm{DPP}}$ and $\lambda_{\mathrm{NEM}}$ on SVHN-semi-1k (reporting validation accuracy)

| | Autoattack $\ell_\infty$ | | | | Autoattack $\ell_2$ | | |
|---|---|---|---|---|---|---|---|
| $\lambda_{\mathrm{DPP}}$ \ $\lambda_{\mathrm{NEM}}$ | 1 | 2 | 4 | $\lambda_{\mathrm{DPP}}$ \ $\lambda_{\mathrm{NEM}}$ | 1 | 2 | 4 |
| 0.25 | 24.42 | 22.46 | 7.70 | 0.25 | 44.38 | 41.56 | 24.38 |
| 0.5 | 22.82 | 16.53 | 8.41 | 0.5 | 42.62 | 41.21 | 24.15 |
| 1 | 25.15 | 24.83 | 7.60 | 1 | 44.31 | 42.79 | 21.74 |

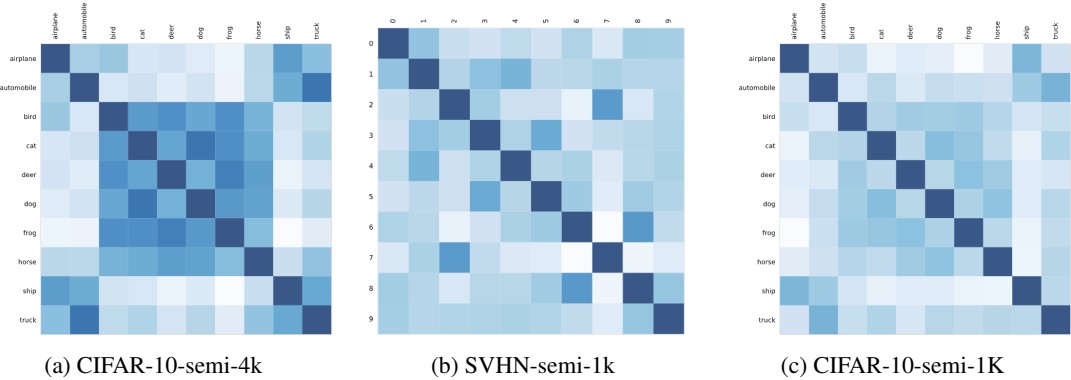

(a) CIFAR-10-semi-4k      (b) SVHN-semi-1k      (c) CIFAR-10-semi-1K

Figure A.5: Visualization of the learned feature-based kernels.

# B   SUPPLEMENTARY RESULTS

**Additional results on CIFAR-10-semi-4k (Table B.6):** In this table, we provide the numerical results that are plotted in Figure 3.

**Additional results on SVHN-semi-1k (Table B.7, Figure B.6):** Here, we evaluate the robustness of ARMOURED variants and SSAR benchmarks against AutoAttack with varying perturbation budgets. The results show that MT+AT achieves the best robustness. Among ARMOURED variants, ARMOURED-H+AT and ARMOURED-F+AT are the most robust and are comparable to each other.

**Additional results from ablation study (Table B.8):** In this table, we report the full evaluation results from our ablation study, adding results from standard attacks. In addition, we create a new model **w/** $H(\mathbf{f})$ by replacing $H(\tilde{\mathbf{f}}_{\backslash y})$ in $\mathcal{L}_{\mathrm{NEM}}$ by the entropy of the averaged prediction $\mathbf{f}$ over all classes, similar to the term used by Pang et al. (2019). This model is less robust than ARMOURED-F+AT against AutoAttack, suggesting that our entropy regularization is better.

**Check on gradient masking (Table B.9):** We evaluate ARMOURED-F, ARMOURED-F+AT and other benchmarks against individual components of AutoAttack. The results show that both

Table B.6: Benchmark against AtutoAttack with varying budgets on CIFAR-10-semi-4k

| Method | AutoAttack $\ell_\infty$ | | | AutoAttack $\ell_2$ | | |
|---|---|---|---|---|---|---|
| | 1/255 | 2/255 | 4/255 | 0.1 | 0.2 | 0.3 |
| RST | $54.38 \pm 0.30$ | $50.80 \pm 0.32$ | $41.35 \pm 0.14$ | $55.04 \pm 0.24$ | $51.34 \pm 0.37$ | $47.44 \pm 0.39$ |
| ARG | $62.57 \pm 0.23$ | $57.83 \pm 0.36$ | $48.09 \pm 1.05$ | $62.27 \pm 0.14$ | $57.11 \pm 0.20$ | $51.83 \pm 0.53$ |
| MT+AT | $59.26 \pm 0.55$ | $53.94 \pm 1.13$ | $43.12 \pm 1.99$ | $59.80 \pm 0.46$ | $54.91 \pm 0.73$ | $49.86 \pm 1.21$ |
| ARMOURED-F | $\mathbf{70.72 \pm 0.88}$ | $57.36 \pm 1.59$ | $33.52 \pm 1.93$ | $\mathbf{71.93 \pm 0.65}$ | $59.40 \pm 0.93$ | $46.45 \pm 1.85$ |
| ARMOURED-I+AT | $65.26 \pm 1.07$ | $60.43 \pm 1.42$ | $48.40 \pm 2.80$ | $65.95 \pm 0.89$ | $61.43 \pm 1.60$ | $55.53 \pm 2.00$ |
| ARMOURED-H+AT | $65.20 \pm 1.39$ | $60.36 \pm 2.64$ | $49.79 \pm 5.31$ | $65.72 \pm 1.11$ | $61.61 \pm 2.70$ | $56.14 \pm 3.53$ |
| ARMOURED-F+AT | $65.75 \pm 2.01$ | $\mathbf{62.14 \pm 2.33}$ | $\mathbf{52.99 \pm 3.53}$ | $65.98 \pm 1.90$ | $\mathbf{62.94 \pm 2.34}$ | $\mathbf{58.79 \pm 2.90}$ |

Table B.7: Benchmark against AtutoAttack with varying budgets on SVHN-semi-1k

| Method | AutoAttack $\ell_\infty$ | | | AutoAttack $\ell_2$ | | |
|---|---|---|---|---|---|---|
| | 1/255 | 2/255 | 4/255 | 0.1 | 0.2 | 0.3 |
| RST | $50.23 \pm 1.88$ | $44.76 \pm 2.07$ | $34.25 \pm 2.08$ | $49.74 \pm 2.26$ | $43.65 \pm 2.77$ | $37.33 \pm 3.16$ |
| ARG | $88.25 \pm 0.32$ | $83.52 \pm 0.21$ | $70.55 \pm 0.72$ | $85.99 \pm 0.31$ | $76.37 \pm 0.76$ | $62.31 \pm 1.22$ |
| MT+AT | $\mathbf{89.37 \pm 0.33}$ | $\mathbf{84.53 \pm 0.55}$ | $\mathbf{71.42 \pm 0.70}$ | $\mathbf{87.99 \pm 0.45}$ | $\mathbf{80.51 \pm 0.65}$ | $\mathbf{69.26 \pm 1.04}$ |
| ARMOURED-F | $81.54 \pm 1.46$ | $64.45 \pm 2.08$ | $33.68 \pm 2.07$ | $82.57 \pm 1.14$ | $66.33 \pm 1.31$ | $49.34 \pm 1.18$ |
| ARMOURED-I+AT | $83.55 \pm 0.21$ | $73.85 \pm 1.83$ | $51.96 \pm 2.96$ | $83.80 \pm 0.07$ | $74.17 \pm 1.33$ | $63.09 \pm 1.98$ |
| ARMOURED-H+AT | $84.58 \pm 0.34$ | $75.37 \pm 0.98$ | $55.49 \pm 1.87$ | $85.09 \pm 0.17$ | $76.25 \pm 0.66$ | $66.27 \pm 1.40$ |
| ARMOURED-F+AT | $84.78 \pm 1.10$ | $75.59 \pm 2.03$ | $54.64 \pm 3.42$ | $85.11 \pm 1.09$ | $76.24 \pm 1.85$ | $65.71 \pm 2.53$ |

ARMOURED-F and ARMOURED-F+AT are very robust against black-box attacks (FAB and Square), which suggests that gradient-masking is less likely to exist in our models.

**Utilization of unlabeled data (Table B.10):** We define the utilization rate as the ratio between the number of stable samples and the total number of unlabeled samples in each minibatch. For each setup, we report the average utilization rate in the last 1000 training iterations. While the utilization rates on CIFAR-10-semi-4k are high (about 90%), they are much lower on CIFAR-10-semi-1k (65%-80%) and SVHN-semi-1k (about 80% except for ARMOURED-F). We suspect that the low utilization rates negatively affect the performance of ARMOURED but was not able to conduct further investigation on this issue.

**Regularization effect of DPP kernel (Figure B.7):** We illustrate the average prediction of test samples generated by ARMOURED variants. From all the subplots, we can clearly see that each network has developed a preference on high or low posterior for each class. For example, in Figure B.7b, network $N^2$ (right side) tends to have high predictions for airplane, automobile, ship and truck, while network $N^1$ (left side) has higher predictions on the remaining six classes. This behaviour is promoted by the hand-crafted kernel. The feature-based kernel (Figure B.7c and Figure B.7d) encourages a similar grouping of classes, even though the distinctions are less severe. With the identity matrix as kernel, the predictions in Figure B.7a also form two groups, but the correlation among classes of the same group are less intuitive.

**Results on CIFAR-10-semi-1k (Table B.11, Table B.12, Figure B.8):** We conduct an experiment on CIFAR-10-semi-1k setup. The results in Table B.11 show that ARMOURED variants achieve higher clean accuracy and better robustness against standard attacks, when compared to SSAR benchmarks. However, under AutoAttack, the best benchmark (ARG for $\ell_\infty$ and MT+AT for $\ell_2$) outperforms ARMOURED-F+AT by about 3%. We suspect that the drops in the performance of ARMOURED are due to low utilization rate (see Table B.10), but were not able to investigate this issue further. In addition, we plot the robustness against AutoAttack for varying perturbation budgets in Figure B.8. Similar to the results on CIFAR-10-semi-4k, ARMOURED-F shows better performance than SSAR benchmarks under small $\epsilon$ attacks.

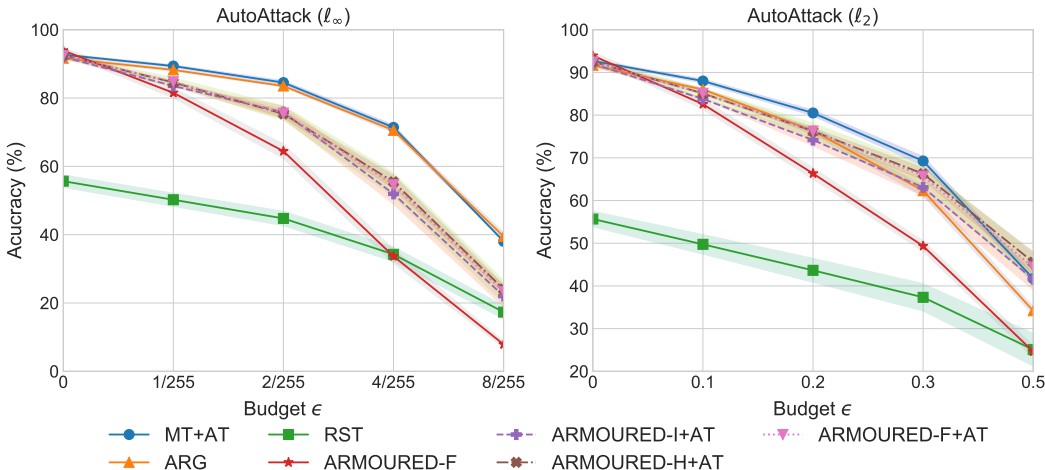

Figure B.6: Robustness against AutoAttack vs. perturbation budget $\epsilon$ on SVHN-semi-1k.

Table B.8: Ablation study on CIFAR-10-semi-4k (full results)

| Method | Clean | FGSM $\ell_\infty$ | PGD $\ell_\infty$ | PGD $\ell_2$ | AutoAttack $\ell_\infty$ | AutoAttack $\ell_2$ |
|---|---|---|---|---|---|---|
| w/o ($\mathcal{L}_{\text{DPP}} + \mathcal{L}_{\text{NEM}}$) | $75.60 \pm 0.44$ | $42.19 \pm 0.40$ | $18.96 \pm 0.22$ | $40.05 \pm 0.91$ | $9.27 \pm 0.28$ | $27.32 \pm 0.85$ |
| w/o $\mathcal{L}_{\text{DPP}}$ | $77.44 \pm 0.85$ | $66.42 \pm 1.27$ | $\mathbf{59.65 \pm 2.07}$ | $65.18 \pm 1.14$ | $34.02 \pm 2.80$ | $48.52 \pm 1.79$ |
| w/ $H(\mathbf{f})$ | $76.91 \pm 0.71$ | $63.81 \pm 1.99$ | $55.83 \pm 3.70$ | $61.93 \pm 2.40$ | $31.20 \pm 6.22$ | $45.07 \pm 6.48$ |
| w/o Unlabeled | $72.50 \pm 0.21$ | $41.66 \pm 1.29$ | $22.26 \pm 2.54$ | $36.29 \pm 1.42$ | $10.26 \pm 1.57$ | $25.23 \pm 0.59$ |
| ARMOURED-F | $\mathbf{84.90 \pm 0.22}$ | $\mathbf{68.27 \pm 1.53}$ | $56.42 \pm 3.93$ | $\mathbf{67.45 \pm 2.76}$ | $8.70 \pm 1.27$ | $25.80 \pm 2.50$ |
| ARMOURED-F+AT | $76.76 \pm 1.60$ | $64.05 \pm 3.00$ | $55.12 \pm 4.90$ | $61.93 \pm 3.06$ | $\mathbf{35.24 \pm 4.56}$ | $\mathbf{49.78 \pm 3.80}$ |

Table B.9: Benchmark against components of AutoAttack on CIFAR-10-semi-4k

| | Attack ($\epsilon$) | MT+AT | ARG | RST | ARMOURED-F | ARMOURED-F+AT |
|---|---|---|---|---|---|---|
| | Clean | $64.77 \pm 0.28$ | $67.07 \pm 0.22$ | $58.81 \pm 0.28$ | $\mathbf{84.90 \pm 0.22}$ | $76.76 \pm 1.60$ |
| $\ell_\infty$ | APGD (8/255) | $24.68 \pm 3.03$ | $30.98 \pm 1.78$ | $30.66 \pm 0.38$ | $20.24 \pm 2.57$ | $\mathbf{44.51 \pm 3.97}$ |
| | FAB-t (8/255) | $24.63 \pm 2.95$ | $30.34 \pm 1.81$ | $25.62 \pm 0.18$ | $44.17 \pm 0.82$ | $\mathbf{62.60 \pm 2.25}$ |
| | Square (8/255) | $29.84 \pm 2.94$ | $35.37 \pm 1.68$ | $29.00 \pm 0.22$ | $\mathbf{77.90 \pm 0.59}$ | $70.99 \pm 1.37$ |
| | AutoAttack (8/255) | $24.28 \pm 2.90$ | $30.01 \pm 1.85$ | $25.38 \pm 0.15$ | $8.70 \pm 1.27$ | $\mathbf{35.24 \pm 4.56}$ |
| $\ell_2$ | APGD (0.5) | $40.48 \pm 1.72$ | $41.71 \pm 0.66$ | $43.05 \pm 0.41$ | $41.25 \pm 2.76$ | $\mathbf{57.62 \pm 2.88}$ |
| | FAB-t (0.5) | $40.13 \pm 1.63$ | $41.15 \pm 0.70$ | $39.94 \pm 0.56$ | $50.14 \pm 1.29$ | $\mathbf{65.69 \pm 1.75}$ |
| | Square (0.5) | $51.87 \pm 1.12$ | $54.26 \pm 0.58$ | $48.76 \pm 0.26$ | $\mathbf{82.05 \pm 0.46}$ | $72.49 \pm 1.32$ |
| | AutoAttack (0.5) | $39.93 \pm 1.63$ | $40.86 \pm 0.69$ | $39.79 \pm 0.60$ | $25.80 \pm 2.50$ | $\mathbf{49.78 \pm 3.80}$ |

Table B.10: Utilization rate of unlabeled data

| | CIFAR-10-semi-4k | SVHN-semi-1k | CIFAR-10-semi-1k |
|---|---|---|---|
| ARMOURED-I+AT | $87.97 \pm 0.86$ | $83.79 \pm 6.94$ | $81.93 \pm 7.04$ |
| ARMOURED-H+AT | $88.61 \pm 0.92$ | $81.93 \pm 1.20$ | $64.99 \pm 2.33$ |
| ARMOURED-F+AT | $87.95 \pm 1.01$ | $82.28 \pm 0.58$ | $65.84 \pm 6.13$ |
| ARMOURED-F | $96.55 \pm 1.22$ | $94.35 \pm 0.47$ | $78.08 \pm 2.08$ |

Table B.11: Benchmark results on CIFAR-10-semi-1k

| Method | Clean | FGSM $\ell_\infty$ | PGD $\ell_\infty$ | PGD $\ell_2$ | AutoAttack $\ell_\infty$ | AutoAttack $\ell_2$ |
|---|---|---|---|---|---|---|
| MT | $65.12 \pm 2.69$ | $4.40 \pm 0.16$ | $0.35 \pm 0.31$ | $9.21 \pm 2.43$ | $0.06 \pm 0.06$ | $3.98 \pm 2.15$ |
| MixMatch | $\mathbf{75.70 \pm 1.35}$ | $16.99 \pm 1.96$ | $0.00 \pm 0.00$ | $0.85 \pm 0.32$ | $0.00 \pm 0.00$ | $0.02 \pm 0.02$ |
| RST | $45.74 \pm 2.07$ | $21.31 \pm 0.74$ | $19.33 \pm 1.17$ | $32.05 \pm 2.38$ | $14.28 \pm 1.78$ | $28.29 \pm 3.08$ |
| ARG | $51.20 \pm 0.67$ | $23.57 \pm 0.55$ | $20.49 \pm 0.79$ | $32.28 \pm 1.11$ | $\mathbf{18.84 \pm 0.99}$ | $30.55 \pm 1.03$ |
| MT+AT | $50.89 \pm 0.37$ | $20.39 \pm 2.03$ | $17.44 \pm 1.96$ | $32.11 \pm 1.30$ | $16.63 \pm 1.83$ | $\mathbf{31.21 \pm 1.32}$ |
| ARMOURED-F | $62.64 \pm 0.48$ | $\mathbf{42.16 \pm 4.64}$ | $\mathbf{31.94 \pm 6.27}$ | $\mathbf{40.20 \pm 4.44}$ | $9.40 \pm 1.51$ | $22.93 \pm 0.75$ |
| ARMOURED-I+AT | $56.35 \pm 1.69$ | $27.31 \pm 4.17$ | $15.73 \pm 3.48$ | $29.16 \pm 2.83$ | $9.59 \pm 2.61$ | $22.40 \pm 2.67$ |
| ARMOURED-H+AT | $54.26 \pm 1.18$ | $26.26 \pm 4.33$ | $13.55 \pm 6.38$ | $26.14 \pm 6.32$ | $8.27 \pm 4.91$ | $19.62 \pm 6.13$ |
| ARMOURED-F+AT | $56.74 \pm 0.97$ | $35.37 \pm 6.34$ | $23.90 \pm 5.99$ | $34.91 \pm 4.41$ | $15.10 \pm 2.72$ | $28.31 \pm 3.75$ |

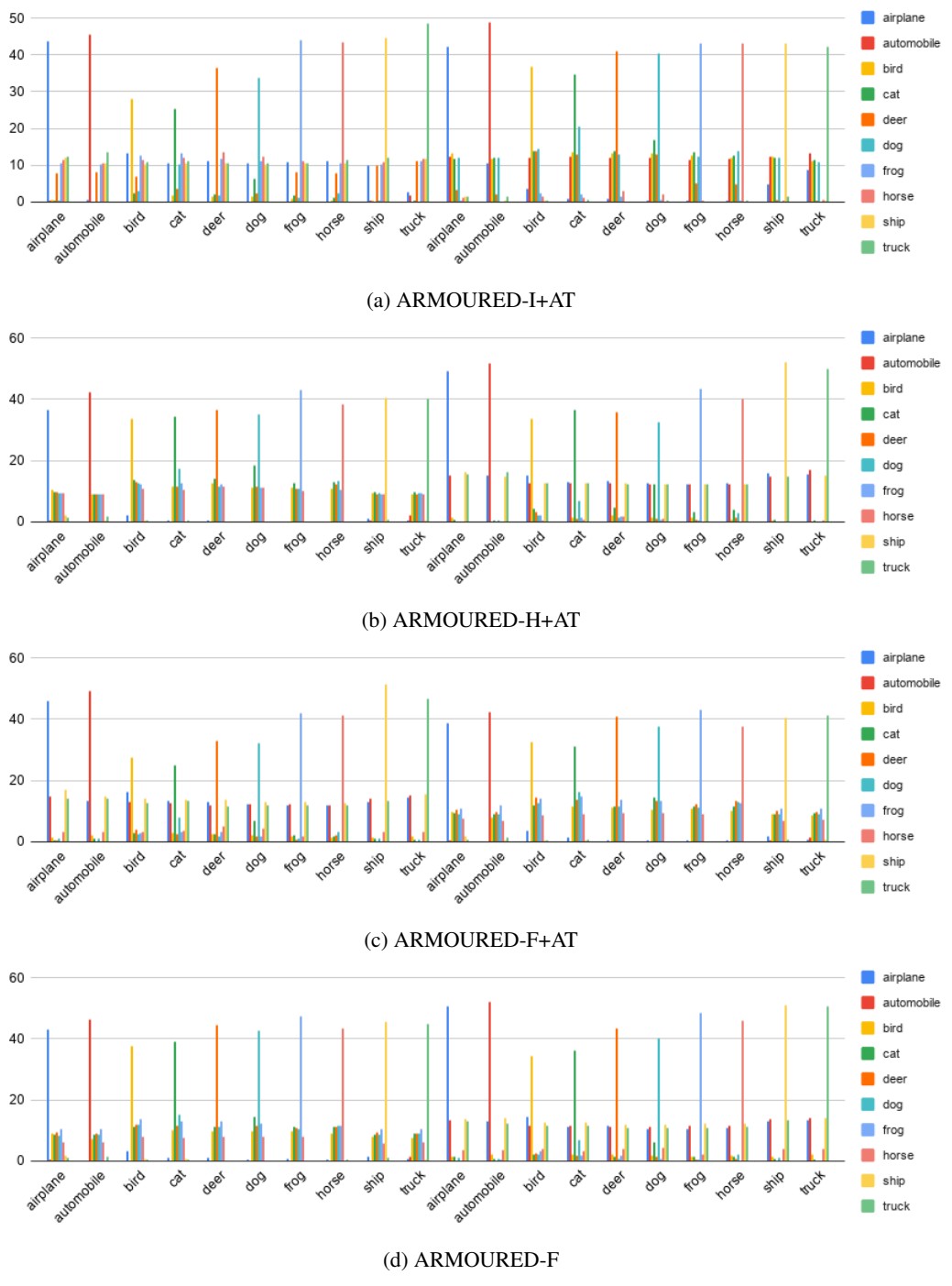

(a) ARMOURED-I+AT

(b) ARMOURED-H+AT

(c) ARMOURED-F+AT

(d) ARMOURED-F

Figure B.7: Posterior outputs on CIFAR-10 test data, generated by networks $N^1$ (left side) and $N^2$ (right side). Horizontal labels show the ground truth classes, vertical axis shows the predicted probabilities multiplied by 100. All models are trained on CIFAR-10-semi-4k.

Table B.12: Benchmark against AtutoAttack with varying budgets on CIFAR-10-semi-1k

| Method | AutoAttack $\ell_\infty$ | | | AutoAttack $\ell_2$ | | |
|---|---|---|---|---|---|---|
| | 1/255 | 2/255 | 4/255 | 0.1 | 0.2 | 0.3 |
| RST | $40.89 \pm 2.12$ | $36.30 \pm 1.89$ | $27.46 \pm 2.01$ | $42.06 \pm 2.37$ | $38.54 \pm 2.27$ | $35.06 \pm 3.54$ |
| ARG | $46.50 \pm 0.75$ | $\mathbf{41.80 \pm 0.85}$ | $\mathbf{33.12 \pm 0.75}$ | $46.89 \pm 0.59$ | $42.57 \pm 0.85$ | $38.58 \pm 0.84$ |
| MT+AT | $45.65 \pm 0.75$ | $40.59 \pm 1.22$ | $31.10 \pm 1.56$ | $46.75 \pm 0.64$ | $42.88 \pm 0.84$ | $\mathbf{38.87 \pm 1.19}$ |
| ARMOURED-F | $\mathbf{48.63 \pm 1.35}$ | $41.44 \pm 1.29$ | $26.57 \pm 0.96$ | $\mathbf{49.43 \pm 1.55}$ | $\mathbf{42.93 \pm 1.17}$ | $35.83 \pm 0.86$ |
| ARMOURED-I+AT | $39.19 \pm 2.63$ | $33.97 \pm 2.92$ | $23.17 \pm 3.31$ | $39.93 \pm 2.41$ | $35.54 \pm 2.62$ | $30.90 \pm 2.89$ |
| ARMOURED-H+AT | $35.98 \pm 3.41$ | $30.60 \pm 4.87$ | $20.35 \pm 6.15$ | $36.75 \pm 3.12$ | $32.41 \pm 4.36$ | $27.52 \pm 5.18$ |
| ARMOURED-F+AT | $42.21 \pm 2.31$ | $38.11 \pm 3.40$ | $29.34 \pm 4.04$ | $42.47 \pm 2.18$ | $39.39 \pm 2.76$ | $35.42 \pm 3.64$ |

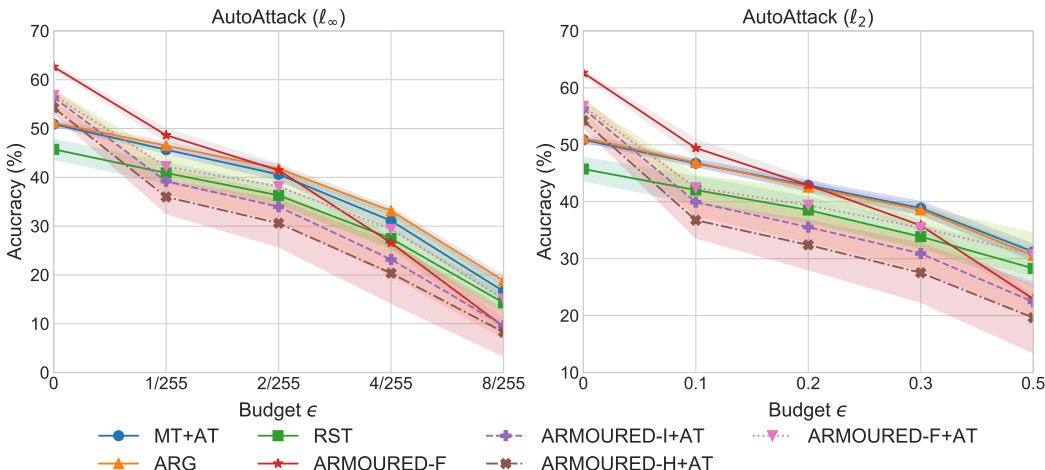

Figure B.8: Robustness against AutoAttack vs. perturbation budget $\epsilon$ on CIFAR-10-semi-1k.

