# OpenReview forum: "ARMOURED: Adversarially Robust MOdels using Unlabeled data by REgularizing Diversity"
_ICLR.cc/2021/Conference — ICLR 2021 Poster_

### Official Review · AnonReviewer2 · 2020-10-27
**A combination of ADP and multi-view training**

**Rating:** 7
**Confidence:** 4

**Review:**

This paper considers training an adversarially robust model in a semi-supervised setting. The authors propose an ensemble-based algorithm for this goal. The algorithm uses a regularization term to induce diversity in the ensemble. The algorithm also leverages the idea of multi-view training in semi-supervised learning to make use of unlabeled data. The experimental results show that the proposed algorithm outperforms several baselines.

Overall I enjoyed reading this paper. It is relatively clearly written, and easy to follow. I think the basic idea of this paper is a combination of the ADP algorithm by Pang et al and the classic multi-view idea in semi-supervised learning. In fact, some of the figures in this paper are quite similar to those in Pang et al 2019. It is not surprising that this algorithm outperforms ADP since it makes use of many unlabeled data. This makes me feel that the proposed algorithm does not have enough novelty. However, I do think this paper is beneficial to the adversarial robustness (AR) community since it brings an important idea of multi-view semi-supervised learning to AR research. Therefore, I decided to give a score of weakly accept.

======
After author response: I have read other reviews and the revised version. I think the paper's overall quality has improved. I decided to change my score to 7.

---

> ### Author Response · Authors · 2020-11-20
> **Thank you**
>
> We thank the reviewer for your feedback.

---

### Official Review · AnonReviewer3 · 2020-10-27
**Interesting techniques and findings, but needs stronger adversarial evaluation**

**Rating:** 7
**Confidence:** 5

**Review:**

**Update** : Since most of my issues have been addressed, I have changed my rating from 6 to 7


Summary:

This paper proposed ARMOURED, a semi-supervised training algorithm that combines multi-view learning to utilize unlabeled data, along with diversity regularization to decrease inter-model transferability for adversarial inputs. As a consequence, the final models achieve near state-of-the-art adversarial robustness, without explicitly generating them during its training process. Based on experiments on the CIFAR10 and SVHN classification tasks, the proposed technique produces models that maintain accuracy on clean as well as adversarially perturbed data.

##########################################################################

Reasons for score:

The proposed regularization terms of using unlabeled data and an ensemble of models with different views to indirectly promote adversarial robustness are quite interesting. However, the absence of evaluation against an adaptive adversary, along with signs of gradient masking makes this work in its current form not ready for publication. If the authors can run tests with an appropriately designed adaptive adversary, along with stronger evaluations (including gradient-free and black-box adversaries), and if the model performs well under those settings, the proposed work would be qualified to be part of the conference proceedings.

##########################################################################

Pros:

- The idea of increasing inter-model diversity to decrease transferability between models is a very interesting concept. The fact that it leads to better robustness without explicitly training for it shows that the authors' intuition in using this regularization term is, in fact, sound.

- Using the idea that augmented images are distorted views of the same image and thus their deep views should agree is great! However, I must point out that the idea of using augmentations to enforce consistency on unlabeled data is now new. One example of such work is [Unsupervised Data Augmentation for Consistency Training](https://arxiv.org/pdf/1904.12848.pdf). Please cite this (and/or relevant work) and do include a comparison.

- Using the concept of a stable sample (based on consensus) is great, since it might help filter out noisy labels.

##########################################################################

Cons:

- Since this is primarily a paper that focuses on adversarial robustness (ultimately), I feel some very basic yet important changes are required. In general, the authors could follow the advice in the paper [On Evaluating Adversarial Robustness](https://arxiv.org/pdf/1902.06705.pdf). To be a bit more concrete, I would recommend having at least the following:
  - Using FGSM to evaluate is not recommended, since it is not nearly as strong an attack as something like PGD. Instead, I would recommend swapping FGSM evaluation with a black-box attack like Boundary++, or some perturbation-minimizing attack like C&W
  - Since the final model is trained using a combination of loss functions and non-trivial processes, it might be beneficial to run evaluations against an adaptive adversary. Space could be made for it by shifting to Figure 3 to supplementary material.

- Is there any particular reason for skipping augmentation at inference time?

- The part that involves using proper class-level information to incorporate it in the DPP regularizer seems similar to work that uses cost-functions to differentiate between different classes when working with adversarial inputs. One such work is [Cost-Sensitive Robustness against Adversarial Examples](https://arxiv.org/abs/1810.09225). It might be worthwhile to talk about and compare the matrices used in the experiments with the matrices used in this work.

- Although the models do theoretically possess similar learning capabilities, it might be interesting to see if different models end up being more "contributing" than others in their predictions. Maybe some analysis might reveal that some augmentations/models are much more useful than the rest (which could be used for pruning), or perhaps trends like 'some models help with clean accuracy, while others help with robustness'. If space/time permits, it would be nice to have such analyses (perhaps in additional materials).

- It has been observed in the literature that the performance of PGD attacks is significantly degraded when random restarts are not used. In your experiments, please run them with random-restarts and report numbers.

- Tables 1&2 have '+-' numbers with them. Why are they there? Are these numbers accuracy, or error rates? Were the attacks run multiple times? If yes, how many? Also, are those numbers s.d? Please clarify.

- The use of unlabeled data along in this case, along with multiple models to have an ensemble effect, seems to be somewhat related to this work [A2-LINK](https://ieeexplore.ieee.org/document/9104705). The authors might benefit from looking at similarities in the algorithm, and if some techniques can be borrowed/adapted to make the current algorithm stronger in its performance.

- The fact that even under strong PGD attacks classes remain recognizable, along with almost unscathed performance for ARMOURED-B for increasing perturbation budgets, strong hints at some form of gradient masking in the final model. Please use black-box evaluation and other methods to make sure this isn't the case. One simple check would be to use [Reliable evaluation of adversarial robustness with an ensemble of diverse parameter-free attacks](https://arxiv.org/abs/2003.01690). Their code is publicly available [here](https://github.com/fra31/auto-attack)

##########################################################################

Typos/Minor edits:

- Abstract, last line "substantial gains in accuracy, while maintaining high accuracy...". Did the authors mean "substantial gains in robustness..."?

- Introduction, third line "..in the input image..". Since the definition being given is for adversarial attacks in general, it should be for any data points with human-imperceptible changes, not just images. Also, in the more broad sense, the examples lead to unintended/unexpected behavior, not necessarily "incorrect classifications".

- Figure 1: Please do not rely on color to differentiate between different arrows (the reader may be color blind). Either switch to some other attribute (like the width of the arrow) or explicitly label them to differentiate between red and blue.

- Formatting: Table 2 could be shifted elsewhere (with text wraparound) to get more space for the authors to focus on things like adaptive adversaries. The same comment for Figure 1 (a lot of extra space on both sides of the image); could remove the padding?

Please address and clarify the cons above. Most importantly, it is crucial to have evaluation against stronger attacks as well as an adaptive adversary, before claims of robustness can be substantiated.

---

> ### Author Response · Authors · 2020-11-20
> **Updates and Further Discussions**
>
> We thank the reviewer for the highly valuable review. With this revision (rev_1), we would like to address the following issues:
>
> > ... it might be beneficial to run evaluations against an adaptive adversary. ... strong hints at some form of gradient masking in the final model. ... Most importantly, it is crucial to have evaluation against stronger attacks as well as an adaptive adversary, before claims of robustness can be substantiated.
>
> We have evaluated the proposed method against stronger adversaries, including Auto-PGD, FAB, Square, and AutoAttack. Please refer to subsection 4.3 of this revision (rev_1), note that the ARMOURED method now includes random augmentation during inference. Experiment results in Table 4 show that ARMOURED-F is robust against FAB and Square, which suggests that gradient-masking is less likely to exists in the final model. ARMOURED-F also yield the highest clean accuracy among all competing methods. However, it is much more vulnerable to APGD, and especially AutoAttack.
>
> On a side note, we found that the original implementation of ARMOURED-F (without augmentation during inference) is robust to $\ell_2$ AutoAttack for small epsilon budgets. For $\ell_2$ attacks with $\epsilon = 0.1$, original ARMOURED-F achieves 74.12%, 64.07%, 60.18% and 60.18% robust accuracies against the sequence of attacks APGD-CE, APGD-T, FAB-T then Square (the AutoAttack sequence). Meanwhile, ARG achieves 62.56%, 62.18%, 62.18% and 62.18% robust accuracies against the same sequence of attacks. While achieving comparable robustness, the clean accuracy of ARMOURED-F is significantly higher at 86.06% versus 66.82% for ARG.
>
> Therefore, we believe that ARMOURED-F alone nonetheless confers some amount of robustness while significantly improving accuracy on clean samples. We further notice that multi-view diversity is an orthogonal approach to adversarial training (AT), and that stronger defenses always rely on AT. Thus, we decide to combine ARMOURED with AT. We propose a variant of ARMOURED-F with adversarial training, namely ARMOURED-F+AT. This model shows strong resilience against all adversaries: its performance asymptotes the state-of-the-art robustness against AutoAttack, and it substantially outperforms other benchmarks against APGD, FAB and Square attacks.
>
> We are thinking about developing this new ARMOURED with Adversarial Training concept as one of the main contributions. We also understand that such a fundamental change at this point may give rise to great uncertainties. Thus, we would like to seek the reviewer’s advice regarding this matter.
>
> > Is there any particular reason for skipping augmentation at inference time?
>
> We thank the reviewer for raising this question. We have made changes to the ARMOURED method to include random augmentation during inference. This change has improved the performance of ARMOURED and has been updated in Tables 1, 2 and 3 of rev_1.
>
> > The part that involves using proper class-level information to incorporate it in the DPP regularizer seems similar to work that uses cost-functions to differentiate between different classes when working with adversarial inputs.
>
> The cost-sensitive robustness paper presents an interesting idea to encode adversarial-cost information onto different classes. However, our DPP kernel works by encoding similarity information between classes, and may not be directly comparable to cost matrices. We still thank the reviewer for bringing this up.
>
> > In your experiments, please run them with random-restarts and report numbers.
>
> We are implementing multiple restarts for PGD attacks. We will update our results accordingly in a future revision.
>
> > Tables 1&2 have '+-' numbers with them. ... Please clarify.
>
> Each result in our tables include the mean and standard deviation over three runs with different random seeds for selecting labeled samples.
>
> > The use of unlabeled data along in this case, along with multiple models to have an ensemble effect, seems to be somewhat related to this work A2-LINK.
>
> We thank the reviewer for this suggestion. The A2-LINK method applies a Mean Teacher-based framework and uses adversarial training to generate covariates. The ideas presented in this work are promising, and will be considered for our future work.
>
> > Typos/Minor edits:
>
> We thank the reviewer for pointing out these mistakes. We have made changes accordingly in this revision (rev_1).

---

### Official Review · AnonReviewer4 · 2020-10-28
**Simple and good technique to increase model robustness during training**

**Rating:** 7
**Confidence:** 5

**Review:**

### **Summary**:

This work introduces ARMOURED, a new method for learning models that are robust against adversarial attacks. The method uses **multi-view**-learning (as in using multiple models with different parameters to cast their votes/views on a given sample), semi-supervised learning to pseudo-label new data based on a consensus, and a diversity regularizer. The approach is evaluated in CIFAR-10 and SVHN against recent baselines in the presence of white-box attacks. The results yield by ARMOURED in these scenarios are superior.

---

### **Reasons for score**:

My recommendation is to accept the paper. I like the proposal, it is simple yet is able to produce better results than existing methods. It also performs well with clean samples, which is something that many other methods struggle with. The only thing I miss is to see experiments in ImageNet.

---
### **Strengths**:

* The problem is relevant for the community and the proposed approach seems to do a better job than the existing state-of-the-art leveraging weak-supervision/semi-supervision (definitely a plus).
* The method is simple and seems easy to adapt to other architectures and tasks.

---

### **Weaknesses**:

* The ablation experiments point to the unlabelled data as one of the core reasons for the method to work. I was expecting to see additional experiments with extra unlabelled data to get a better feeling of this component.

---

### **Questions to be discussed during the rebuttal period**:

Please, address the questions/comments expressed in the **Weaknesses** section.

---

### **Other considerations**:

None


---

---

> ### Author Response · Authors · 2020-11-20
> **Clarifications**
>
> We thank the reviewer for your comments. We would like to address your requests as follows:
>
> > I would love to see results in ImageNet.
>
> Due to our limited time and resources, we are afraid that we may not be able to conduct an evaluation on ImageNet.
>
> > I was expecting to see additional experiments with extra unlabelled data to get a better feeling of this component.
>
> We are also interested in seeing how additional unlabeled data can boost the performance. However, due to other issues, we are currently focusing on improving the robustness of the method against different types of attacks. We hope to address this issue in a future revision.
>
> > The results of Table 3 create confusion on the actual contribution of H(f) to the method. I think it would be beneficial to provide a better explanation of this issue.
>
> The ADP paper actually uses $H(\mathbf{f})$ in combination with the DPP regularizer. Thus, in Table 3, we would like to show the performance gains by switching from $H(\mathbf{f})$ to the new NEM regularizer. As shown in Table 3, the quantitative gain from applying this change is not substantial, but Figure 3 shows clearly the difference between the learned representations from NEM and $H(\mathbf{f})$.

---

### Official Review · AnonReviewer1 · 2020-10-28
**Review: clear paper with solid motivation and experimental evaluation. Some questions about the interpretation of the results.**

**Rating:** 7
**Confidence:** 3

**Review:**

## Summary
This paper introduces a semi-supervised learning procedure that does not require labeled adversarial data to learn an ensemble model that is robust to adversarial attacks on classification tasks.

## Quality
This paper is very well written; the design decisions of the training procedures are all supported by ablation tests and comparisons to other modern adversarial baselines. However, I am slightly worried by the ablation study results -- from Table 3, it seems like the DPP component of the loss does not provide a statistically significant lift.

## Clarity
This paper is very clearly written! The choice of parameterization, experiment settings, and overall proposed methodology were clearly explained and motivated.

## Originality
As the authors have mentioned, improving diversity in ensembles is not a novel approach to improving robustness to adversarial attacks; the originality of the method lies in the parameterization of diversity through a DPP component and multi-view complementarity.

## Significance
Improving robustness to adversarial attacks is in of itself an important contribution to this field. This paper proposes a method that achieves a significant improvement over other competitors (Tables 1, 2); the proposed method is additionally intuitive, and can incorporate prior beliefs over the the data distribution, making it adaptable to several different settings.

## Questions
### Methodology
My main question lies in the uses of the DPP loss: did the authors use the determinant directly? Depending on the choice of kernel, I would expect the determinant portion of the loss to eclipse the other terms, potentially hampering learning.

On a related note, I am curious to know how sensitive the ARMOURED performance is to the $\lambda_{DPP}$ and $\lambda_{NEM}$ hyper-parameters.

### Experiments
Do the authors have insight in the significant gap between ARMOURED-B and its variants for a large $\epsilon$ budget with $L_\infty$ PGD? I am surprised by the significant gap in Table 3.

Again in Table 3, comparing the -F, no DPP, and -B variants, it seems like the highest lift in performance comes from the diversity in the NEM component. Do you know if a similar lift is observed by removing the NEM component and keeping the DPP?

At inference time, you state that the augmentation step is skipped. Naively, I would not have been surprised to see these augmentations improve resilience to adversarial attacks. Am I incorrect?

---

> ### Author Response · Authors · 2020-11-20
> **Clarifications**
>
> We thank the reviewer for the helpful comments. We would like to answer some of your questions with this revision (rev_1). The remaining questions will be addressed in a future revision:
>
> > My main question lies in the uses of the DPP loss: did the authors use the determinant directly?
>
> We thank the reviewer for bringing this up. Our DPP loss never uses the determinant directly. Instead, a log function is applied to the determinant term. This practice was originally proposed by the ADP method. In our original submission, there was a typo in the DPP equation. We have fixed the DPP loss in section 3 of rev_1.
>
> > Do the authors have insight in the significant gap between ARMOURED-B and its variants for a large $\epsilon$  budget with $L_\infty$ PGD?
>
> The high performance of ARMOURED-B on large-epsilon PGD attacks was also questioned by another reviewer and was suspected to gradient-masking. We are investigating this issue and will provide an update in the next revision.
>
> >  Do you know if a similar lift is observed by removing the NEM component and keeping the DPP?
>
> This is a good question. Because there exist trivial solutions to the DPP loss function, removing the NEM component completely will cause numerical issues. This phenomenon has been studied in the ADP paper.
>
> > Naively, I would not have been surprised to see these augmentations improve resilience to adversarial attacks. Am I incorrect?
>
> We thank the reviewer for this suggestion. We have made changes to the ARMOURED method to include random augmentation during inference. This change has improved the performance of ARMOURED and has been updated in Tables 1, 2 and 3 of rev_1.

---

### Author Response · Authors · 2020-11-20
**Revision 1 (rev_1): Major Updates**

We thank the reviewers for the valuable comments and positive feedback. The major updates in Revision 1 (rev_1) are as follows:

* A common question was whether augmentation would help during inference. We have tried adding random augmentation and found that it improves the robustness of ARMOURED. Thus, we have made changes to the ARMOURED inference procedure in the methodology section itself (c1). Another concern is regarding adding random initialization for PGD $\ell_\infty$, because our original PGD $\ell_\infty$ implementation started from the clean sample. We have fixed this issue. Now, both PGD $\ell_\infty$ and PGD $\ell_2$ start from a randomly perturbed sample (c2). Both changes (c1) and (c2) have been applied to create new results for Tables 1, 2 and 3. The overall trends and observations from these results have also been updated.

* A critical request is to evaluate ARMOURED against stronger attacks, including adaptive attacks and black-box attacks. To address this, we conducted a new experiment in subsection 4.3 to evaluate against Auto-PGD (better convergence than PGD), FAB (white-box perturbation-minimization attack), Square (black-box attack) and AutoAttack (ensemble attack). The results in Table 4 suggest that (i) ARMOURED-F achieved highest clean accuracy (by 10%-15% margin), (ii) gradient-masking does not develop in the ARMOURED-F model, evident by its high robustness against FAB and Square, and (iii) when combined with Adversarial Training, ARMOURED-F approaches state-of-the-art robustness against AutoAttack, and outperforms other benchmarks against APGD, FAB and Square attacks.

We are considering revising the paper towards ARMOURED with Adversarial Training and would appreciate any further feedback. Due to time and resource limits, we have prioritized key concerns; nonetheless we will continue to address any further and remaining issues over the rest of the discussion period.

---

> ### Author Response · Authors · 2020-11-25
> **Revision 2**
>
> Once again, we thank the reviewers for the helpful feedback. In this second update, we have further expanded our experiments using stronger adversaries. we believe that the results, summarized below, address concerns about robustness against stronger attacks, gradient masking and proper evaluation.
>
> On CIFAR-10, we found that ARMOURED-F is robust against AutoAttack under small perturbation budgets ($\epsilon \leq 2/255$ for $\ell_\infty$ attacks and $\epsilon \leq 0.2$ for $\ell_2$ attacks) while obtaining distinctively higher clean accuracy than competing methods. Furthermore, when ARMOURED-F is combined with adversarial training (AT), the resulting model is robust against a wider range of perturbation budgets, while nonetheless maintaining higher accuracy on clean data than competing methods. These results shown in Figure 3 of the revised paper suggest that our core contribution of regularizing diversity through multi-view learning has merit even under strong attacks, and can be combined with AT to enable robustness under larger perturbation budgets while maintaining higher accuracy on clean samples.
>
> We have revised the paper replacing the previous experiments using PGD with results from AutoAttack, including an ablation study demonstrating the benefits of the various components in the regime of small perturbation budgets; previous PGD results have been moved to Appendix B. Due to the limited time we are unable to revise all the results (e.g. on SVHN) but will include these in the final version of the paper.

---

### Decision · Program_Chairs · 2021-01-07
**Final Decision**

**Decision:**

Accept (Poster)

**Comment:**

This paper focuses on adversarial robustness with unlabeled data. The philosophy behind sounds quite interesting to me, namely, utilizing unlabeled data to enforce labeling consistency while reducing adversarial transferability among the networks via diversity
regularizers. This philosophy leads to a novel algorithm design I have never seen, i.e., ARMOURED, an adversarially robust training method based on semi-supervised learning.

The clarity and novelty are clearly above the bar of ICLR. While the reviewers had some concerns on the significance, the authors did a particularly good job in their rebuttal. Thus, all of us have agreed to accept this paper for publication! Please carefully address all
comments in the final version.